# Blocking muscle wasting via deletion of the muscle-specific E3 ligase MuRF1 impedes pancreatic tumor growth

Daria Neyroud[1,2,3], Orlando Laitano[2,4], Aneesha Dasgupta[5,6], Christopher Lopez[1,2], Rebecca E. Schmitt[6], Jessica Z. Schneider[6], David W. Hammers[2,7], H. Lee Sweeney[2,7], Glenn A. Walter[2,8], Jason Doles[5,6], Sarah M. Judge[1,2] & Andrew R. Judge [1,2✉]

Cancer-induced muscle wasting reduces quality of life, complicates or precludes cancer treatments, and predicts early mortality. Herein, we investigate the requirement of the muscle-specific E3 ubiquitin ligase, MuRF1, for muscle wasting induced by pancreatic cancer. Murine pancreatic cancer (KPC) cells, or saline, were injected into the pancreas of WT and MuRF1$^{-/-}$ mice, and tissues analyzed throughout tumor progression. KPC tumors induces progressive wasting of skeletal muscle and systemic metabolic reprogramming in WT mice, but not MuRF1$^{-/-}$ mice. KPC tumors from MuRF1$^{-/-}$ mice also grow slower, and show an accumulation of metabolites normally depleted by rapidly growing tumors. Mechanistically, MuRF1 is necessary for the KPC-induced increases in cytoskeletal and muscle contractile protein ubiquitination, and the depression of proteins that support protein synthesis. Together, these data demonstrate that MuRF1 is required for KPC-induced skeletal muscle wasting, whose deletion reprograms the systemic and tumor metabolome and delays tumor growth.

[1] Department of Physical Therapy, University of Florida, Gainesville, FL, USA. [2] Myology Institute, University of Florida, Gainesville, FL, USA. [3] Institute of Sports Sciences, University of Lausanne, Lausanne, Switzerland. [4] Department of Applied Physiology & Kinesiology, University of Florida, Gainesville, FL, USA. [5] Department of Anatomy, Cell Biology and Physiology, Indiana University School of Medicine, Indianapolis, IN, USA. [6] Department of Biochemistry and Molecular Biology, Mayo Clinic, Rochester, MN, USA. [7] Department of Pharmacology and Therapeutics, University of Florida, Gainesville, FL, USA. [8] Department of Physiology and Aging, University of Florida, Gainesville, FL, USA. ✉email: arjudge@phhp.ufl.edu

C achexia is a devastating unmet medical condition that is causative in decreasing the quality of life and survival time of patients with cancer[1]. This condition, defined by an involuntary loss of muscle mass with or without a concomitant loss of fat mass[2], affects up to 80% of cancer patients[3] and may account for 20-30% of all cancer-related deaths[2,4,5]. As a result of the ongoing skeletal muscle wasting, treatment toxicity increases and treatment tolerance and efficacy decrease, thereby compromising cancer treatment options in patients presenting with cachexia[3,6,7].

Broadly, muscle wasting in the context of cancer is caused by factors released from the tumor itself and/or the host in response to the tumor, which can alter the net protein balance by reducing skeletal muscle protein synthesis and increasing skeletal muscle protein degradation[8,9]. With respect to protein degradation, the ubiquitin-proteasome system is a major contributor[10], and evidence of increased protein ubiquitination and proteasomal-mediated protein degradation has been reported in skeletal muscle of cachectic tumor-bearing hosts[8,9,11–16]. Within this pathway, proteins are ubiquitinated via a cascade of enzymatic reactions that involve ubiquitin-activating (E1), ubiquitin-conjugating (E2), and ubiquitin-ligating (E3) enzymes. E3 enzymes confer specificity of the reaction through transferring ubiquitin from the E2 enzyme to a specific protein substrate. In the context of muscle wasting, various E3 ligases have been shown to be involved in contractile protein degradation (see[17–19] for review), including the muscle-specific E3 ligases—F-Box Protein 32 (Fbxo32/atrogin-1)[17,20–22] and muscle RING finger protein 1 (MuRF1/Trim63)[17,20,23–28]—as well as the more ubiquitously expressed E3 ligase, ubiquitin protein ligase E3 component N-Recognin 2 (Ubr2)[21,29].

Of the muscle-specific E3 ligases, MuRF1 has been widely studied and found to be strongly implicated in several muscle wasting conditions[20,23,24,26–28,30,31]. An increase in MuRF1 expression is sufficient to induce muscle wasting[30], while deletion of MuRF1 protects against the loss of muscle mass in response to denervation[20], glucocorticoid treatment[24], amino acid deprivation[32], hindlimb suspension[25] and aging[23], but not in response to microgravity[31]. However, to our knowledge, the requirement of MuRF1 has never been tested in cancer-induced muscle wasting, which represents a completely distinct atrophy condition in that it is the only condition in which atrophy is driven by the presence of a tumor. In support of a potential role for MuRF1 in cancer-induced muscle wasting, the mRNA levels of Trim63 (which encodes the MuRF1 protein), are increased in skeletal muscle in several different pre-clinical models of cancer cachexia[9,16,21,33,34] and, in our hands, are increased in the muscle of cachectic, but not in non-cachectic, patients with pancreatic ductal adenocarcinoma (PDAC)[35]. Moreover, delivery of the MuRF1 inhibitors, Myomed-205 or Myomed-206, was recently shown to inhibit cancer-induced muscle atrophy in a murine melanoma model[26]. Building on these findings of others, and to further explore a mechanistic role of MuRF1 in the skeletal muscle of tumor-bearing hosts, we tested the requirement of MuRF1 for pancreatic cancer-induced skeletal muscle atrophy and weakness, and further aimed to identify the MuRF1 skeletal muscle ubiquitinome during the progression of cancer-induced muscle atrophy. Overall, our results demonstrate that MuRF1 deletion (i) extends survival, (ii) protects against body, skeletal muscle and fat wasting, and against skeletal muscle weakness in response to tumor burden, and (iii) slows tumor growth. Through unbiased omics analyses in skeletal muscle, we further show that the protection afforded through loss of MuRF1 is related to not only the role of MuRF1 in mediating muscle protein ubiquitination and degradation but also, to the depression of pathways that support protein synthesis and disruptions to muscle metabolism. Through these muscle-specific functions, our data suggest that MuRF1 deletion may slow tumor growth through depriving the tumor of key energy substrates necessary for rapid tumor growth.

## Results and discussion

**MuRF1 deletion protects against body, skeletal muscle and fat wasting, and extends survival in response to tumor burden.** To determine the extent to which MuRF1 is required for cancer-associated muscle wasting, we inoculated WT and MuRF1 knockout (MuRF1$^{-/-}$) mice with murine pancreatic cancer (KPC) cells. WT and MuRF1$^{-/-}$ mice were euthanized 16 days after KPC cell injection, which corresponds to the time point at which WT KPC mice reached humane endpoint in our hands. At this time, WT KPC mice showed significant body wasting (−12%), muscle wasting (−8 to 17%), fat wasting (–71%), and splenomegaly (+100%) compared to WT control mice (Fig. 1a–f). At this same time point, MuRF1$^{-/-}$ KPC mice showed markedly smaller tumors (–83%, Fig. 1g), and were spared from body, muscle and fat wasting compared to control MuRF1$^{-/-}$ mice (Fig. 1a–e). Based on these findings, implicating a delay in tumor growth in mice lacking MuRF1, we inoculated additional MuRF1$^{-/-}$ mice with KPC cells (hereafter as MuRF1$^{-/-}$ KPC END) and tracked them until they reached humane endpoint, or had a tumor size comparable to WT KPC mice at endpoint. One mouse reached humane endpoint 21 days following KPC cell injection, while the other mice were tracked for a total of 35–36 days. Although these mice presented a wide range of tumor masses (0.15 g to 3.54 g), the mean tumor mass of MuRF1$^{-/-}$ KPC END (1.5 ± 1.3 g) mice was comparable to that reached by WT KPC mice at END (1.9 ± 0.5 g, Fig. 1g). Despite comparable tumor mass, no body, muscle or fat wasting was observed in MuRF1$^{-/-}$ mice (Fig. 1a–e), demonstrating that MuRF1 is required for KPC-induced catabolism of muscle and fat.

We next determined the extent to which MuRF1 deletion is protective against skeletal muscle atrophy in response to KPC tumor burden by immunostaining tibialis anterior and soleus cross-sections with wheat-germ agglutinin (WGA). Skeletal muscle atrophy was observed in both tibialis anterior and soleus in response to KPC tumors in WT mice (Fig. 1h–m). Tibialis anterior muscle showed a leftward shift in its fiber size distribution (Fig. 1i), whereas the soleus showed a greater variability in its fiber size distribution (Fig. 1l), as shown by an increase in the percentage of small and large fibers compared to control mice. Deletion of MuRF1 prevented both KPC-induced tibialis anterior and soleus muscle fiber atrophy (Fig. 1j&m). We further assessed soleus typology and observed no tumor-induced shift in fiber type composition in either genotype (Fig. 1n, o). Taken together, these results establish that MuRF1 deletion not only slows tumor growth and extends survival, but protects against KPC-induced wasting of muscle and fat, despite controlling for the effects of MuRF1 deletion on delayed tumor growth.

**MuRF1 deletion protects against KPC-induced muscle weakness.** We next performed in vitro analyses of soleus contractility to assess the extent to which the protection conferred by MuRF1 deletion on muscle mass and size translated to muscle function. To increase translational relevance, we chose to study the soleus muscle as its mixed typology (i.e., ~40% type I, ~40% type IIA, 6% type IIX fibers[36]), closely resembles human muscle typology[37]. Peak twitch and maximal tetanic force were lower in solei from WT KPC mice compared to WT sham mice (−23% and −32%, respectively), whereas deletion of MuRF1 prevented these reductions in response to KPC tumors (Fig. 2a.i-d.i and Fig. 2a.ii-d.ii).

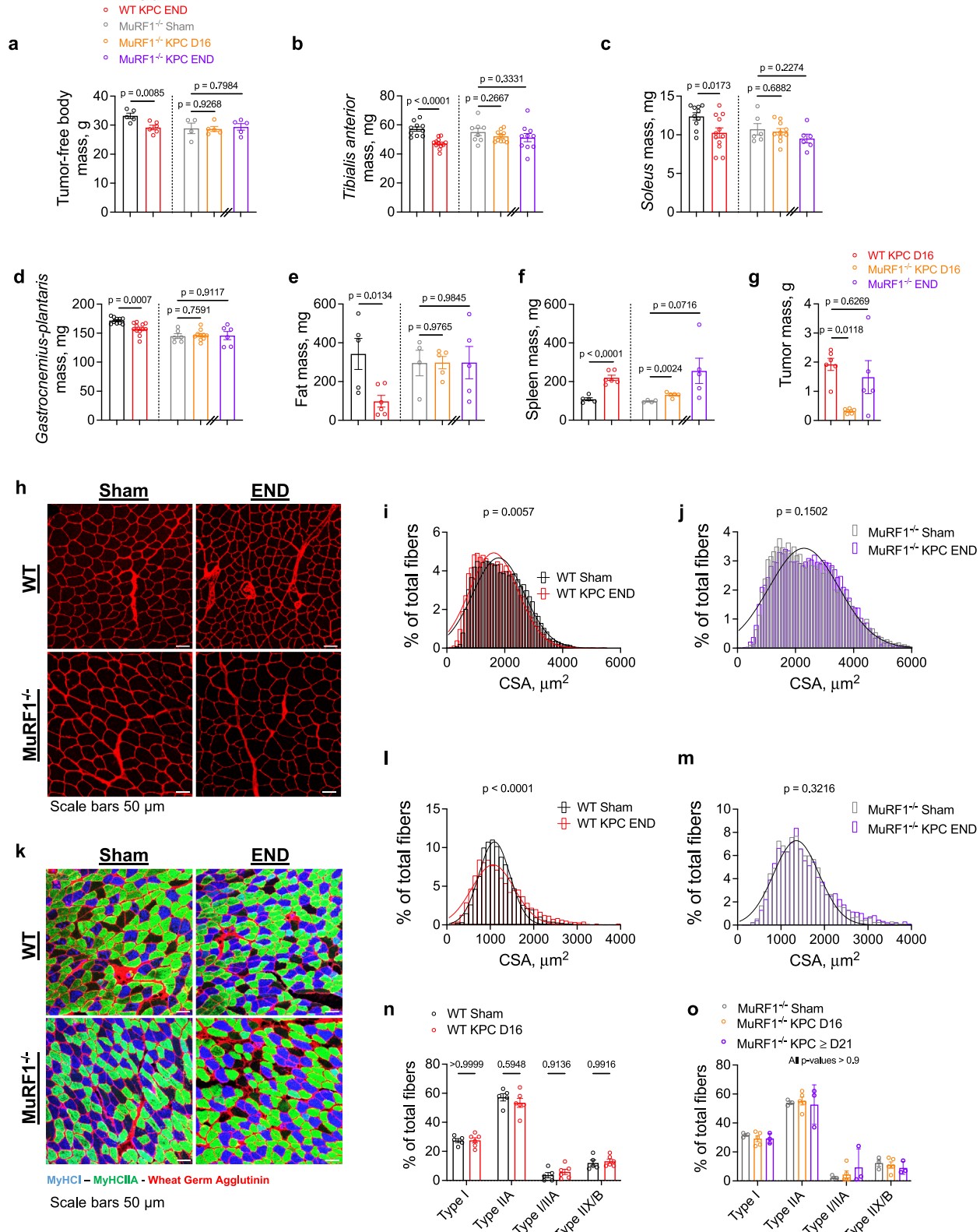

Note that tumor burden did not affect *soleus* muscle relaxation (Fig. 2e, f). To gain further insights into the mechanisms underlying these force reductions, we determined the force-frequency relationship and observed a downward shift in WT KPC mice suggesting a reduced ability of cross-bridges to generate force (Fig. 2g). No such shift was observed in MuRF1⁻/⁻ KPC *solei* (Fig. 2h). To determine if the KPC-induced muscle weakness in

WT mice is associated with increased fatigability, we subjected *solei* to 600 consecutive evoked contractions (200-µs pulse width, 100 Hz, 330-ms duration) and fit the measured evoked forces with an exponential one-phase decay model. While tumor burden led to significant reductions in initial (i.e., $Y_0$ parameter) and asymptotic (i.e., plateau parameter) evoked forces, it did not affect the force decay rate constant (i.e., K parameter) in WT mice suggesting that

**Fig. 1 MuRF1 (*Trim63*) deletion protects against muscle wasting, skeletal muscle fiber atrophy and slows tumor growth. a–f** KPC tumor burden induces reduction in **a** tumor-free body mass ($\eta^2 = 0.5557$), **b–d** skeletal muscle mass ($\eta^2 = 0.6116$ for *tibialis anterior*, $\eta^2 = 0.2520$ for *soleus* and $\eta^2 = 0.4484$ for *gastrocnemius-plantaris*), and **e** fat mass ($\eta^2 = 0.5109$) in WT mice but not MuRF1$^{-/-}$ mice ($\eta^2 < 0.15$), and **f** increase in spleen mass ($\eta^2 = 0.8354$) in WT and MuRF1$^{-/-}$ KPC D16 ($\eta^2 = 0.7542$) mice, while only a trend towards an increased spleen mass was found in MuRF1$^{-/-}$ KPC END mice ($\eta^2 = 0.3913$). **g** MuRF1 deletion slowed tumor mass growth ($\eta^2 = 0.4837$). **h** Representative images of *tibialis anterior* cross-sections stained with wheat-germ agglutinin (WGA). **i, j** Quantification of *tibialis anterior* fiber cross-sectional area reveals that MuRF1 deletion prevents tumor-induced fiber atrophy ($R^2 > 0.9$ for all groups). **k** Representative images of *soleus* cross-sections stained for wheat-germ agglutinin (WGA), myosin heavy chain type I (MHCI), and myosin heavy chain type IIA (MHCIIA). **l, m** Quantification of *soleus* fiber cross-sectional area revealing that MuRF1 deletion prevented increases in the number of small and large fibers in response to tumor burden ($R^2 > 0.9$ for all groups; $n = 3$–6 per group). In **i, j** and **l, m**, cross-sectional area data were binned, fit with a Gaussian least squares regression and significance was determined by calculating the extra sum-of-squares *F*-test. **n, o** Quantification of type I and type IIA percentage indicating that tumor burden did not lead to a fiber type shift in either genotype. Data are presented as means ± SE in panels **a–g** and **n, o**. For panels **i, j**, and **l, m**, $n = 5$ for WT Sham, $n = 6$ for WT KPC END, $n = 4$ for MuRF1$^{-/-}$ Sham and $n = 5$ for MuRF1$^{-/-}$ KPC END.

KPC-tumor burden did not result in greater fatigability (Supplementary Fig. 1a). Similar results were observed in MuRF1$^{-/-}$ mice (Supplementary Fig. 1b).

Given that respiratory failure represents a major cause of deaths in cachectic patients[38], we further determined if MuRF1 deletion protects against KPC-induced diaphragm dysfunction. Similar to our previously published findings[39], strips of diaphragm from WT tumor-bearing mice showed lower peak twitch (-52%) and maximal tetanic force (-46%) compared to WT sham mice, while this cancer-associated diaphragm weakness was prevented by MuRF1 deletion (Fig. 2i–l). KPC tumor burden also increased twitch half-relaxation time in diaphragm strips from WT mice (Fig. 2m), suggesting impaired calcium handling[40]. MuRF1 deletion prevented this slowing of muscle relaxation (Fig. 2n). In addition, a downward shift in the force-frequency relationship was observed in diaphragm from WT KPC mice, whereas no such shift was shown by diaphragm strips from MuRF1$^{-/-}$ mice further supporting that MuRF1 deletion protects against cancer-associated diaphragm weakness (Fig. 2k, l). Overall, our data indicate that KPC-tumor burden results in lower maximal and specific force-generating capacity, and that MuRF1 is causative in this dysfunction.

**MuRF1 mediates cytoskeletal and contractile protein ubiquitination in response to KPC tumor burden.** Given that MuRF1 is a muscle-specific E3-ubiquitin ligase, we next sought to identify proteins in skeletal muscle whose ubiquitination during the initiation and progression of cancer-induced muscle wasting are MuRF1 dependent. To do this, *tibialis anterior* muscles from WT and MuRF1$^{-/-}$ cancer-free and KPC tumor-bearing mice were harvested at different time points throughout the progression of tumor growth, as shown in Fig. 3a–e. Muscles were digested with trypsin and homogenates were immunoenriched for peptides carrying diGly ubiquitin remnant motifs, as previously described[30,41] diGly modified peptides were then identified by label-free LC-MS/MS. Principal component analysis (PCA) revealed a good separation between genotypes, as well as between time points for WT mice (Fig. 4a). Across both genotypes and all time points, a total of 815 ubiquitination sites, mapping to 302 proteins, were detected (Supplementary Data 1). Figure 4b, c depicts the number of ubiquitination sites and the number of proteins whose ubiquitination status changed during tumor growth for both WT and MuRF1$^{-/-}$ mice.

Interestingly, the number of identified sites showing increased ubiquitination almost doubled in skeletal muscle of WT mice between D8 (206 sites), when there is no muscle wasting, and D12 (367 sites), when muscle wasting is first detectable (Fig. 3b–d), and then plateaued. This demonstrates that a large increase in protein ubiquitination parallels the induction of muscle wasting. However, despite an increase in the number of sites showing increased ubiquitination, the total number of proteins showing increased ubiquitination in WT mice remained relatively constant throughout tumor progression (105 at D8 and 198 at END, Fig. 4c). This suggests that some proteins ubiquitinated early during tumor growth may be ubiquitinated on additional sites as the tumor grows and cachexia develops (Fig. 4d). We also identified a significant number of ubiquitination sites and proteins that showed reduced levels of ubiquitination in response to KPC tumors, that similarly increased in number throughout tumor progression. Importantly, while muscles from MuRF1$^{-/-}$ mice also showed early tumor-induced changes in protein and site-specific ubiquitination, these changes remained stable through experimental endpoint, indicating a key role for MuRF1 in regulating the changes in protein ubiquitination during tumor progression and the development of cachexia.

We next performed bioinformatic analyses to identify biological pathways associated with proteins showing increased ubiquitination in the skeletal muscle of WT KPC mice. These analyses revealed an enrichment of proteins related to muscle structure, glycolytic processes and protein proteasomal degradation (Fig. 4e–i). As noted above, increased ubiquitination of skeletal muscle proteins occurs early during tumor growth, prior to muscle wasting, with 64 proteins showing increased ubiquitination of at least one site 8 days following KPC cell inoculation (Fig. 4b, c). Among these 64 proteins, most were related to sarcomeres (including ACTN2, ACTN3, MYH4, TTN), sarcoplasmic reticulum (including ATP2A1, CACNA1S, FKPB1A, JPH2, RYR1) and glycolytic processes (including ALDOA, CKM, GAPDH, PYGM; Fig. 4e and Supplementary Data 2), with many of these proteins also showing increased ubiquitination at later time points. On day 12, when cachexia was first observed, a total of 77 proteins showed increased ubiquitination levels in KPC mice compared to Sham controls. Of these 77 proteins, 50 also showed increase ubiquitination at D10 (Fig. 4f and Supplementary Data 2), and annotated to categories of muscle contraction regulation (including ANXA6, ATP2A1, RYR1, TNNC2, TNNT3, TNNI2), proteasome system (including ADRM1, PSMA2, PSMD4, RAD23, TRIM63/MURF1, VCP), sarcolemma (including ANXA6, ATP2A1, CACNAS1, DES, MYH4, PPP3CA, RYR1), cytoskeleton (including CACNAS1, DES, MYBPC2, MYOM1, TPM1, TPM2, TTN), sarcoplasmic reticulum (including ATP2A1, CACNA1S, JPH2, PPP3CA, RYR1) and gluconeogenesis (including CKM, GAPDH, LDHA, PGAM1, PGAM2, PYGM, TPI1; Fig. 4g and Supplementary Data 2). Similar findings were observed at later stages of cachexia (Fig. 4h, i and Supplementary Data 2).

To further identify proteins whose ubiquitination level increased in tumor-bearing mice through a MuRF1-dependent manner, we overlapped the specific sites, and proteins showing increased ubiquitination in WT KPC mice at each time point with those showing increased ubiquitination at one time point or more

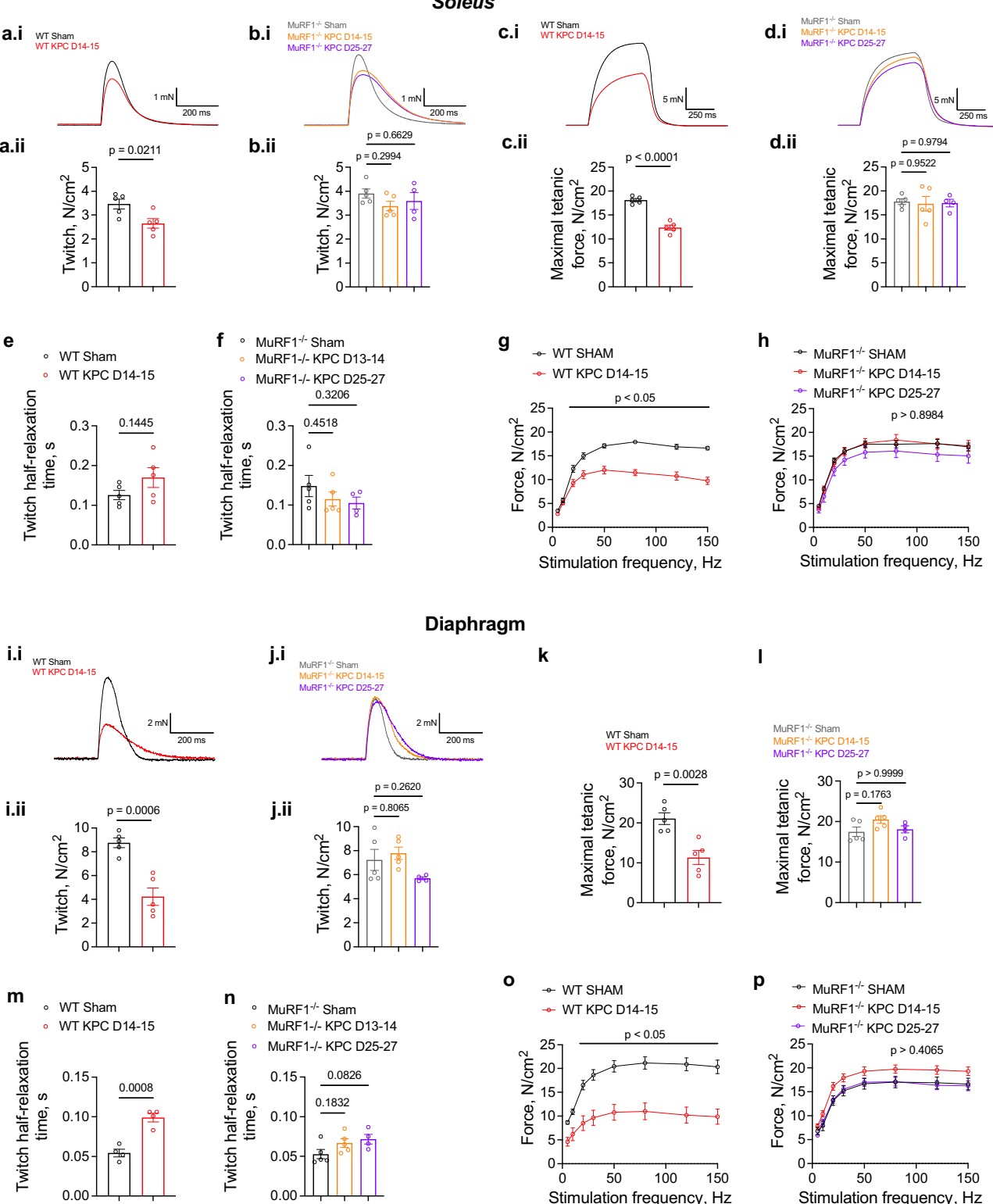

**Fig. 2 MuRF1 deletion protects against of KPC-induced muscle weakness. a–f** KPC tumor burden induces reductions in maximal (**a.i-b.i**) and relative (**a.ii-b.ii**) *soleus* peak twitch ($\eta^2 = 0.5059$), and maximal (**c.i-d.i**) and relative (**c.ii-d.ii**) tetanic force ($\eta^2 = 0.9152$) in WT but not MuRF1$^{-/-}$ mice ($\eta^2 < 0.2$), while it did not affect twitch relaxation rate (**e, f**, $\eta^2 = 0.1842$ in WT mice and 0.1670 in MuRF1$^{-/-}$ mice). Reductions in forces evoked at frequencies >15 Hz (**g, h**) were also observed in response to tumor burden in WT but not MuRF1$^{-/-}$ mice ($\omega^2$ (associated with tumor burden status) = 0.1731 in WT mice and 0.0230 in MuRF1$^{-/-}$ mice; $n = 5$ per group). **i–p** KPC tumor burden induces reductions in diaphragm peak twitch (**i, j**, $\eta^2 = 0.7862$ in WT and 0.3267 in MuRF1$^{-/-}$ mice) and maximal tetanic force (**k, l**, $\eta^2 = 0.6923$ in WT and 0.3239 in MuRF1$^{-/-}$ mice), slows twitch relaxation (**m, n**, $\eta^2 = 0.8639$ in WT and 0.3401 in MuRF1$^{-/-}$ mice) and leads to reduced forces in response to stimulations delivered at frequencies >15 Hz (**o, p**) in WT but not MuRF1$^{-/-}$ mice ($\omega^2$ (associated with tumor burden status) = 0.4691 in WT mice and 0.0674 in MuRF1$^{-/-}$ mice; $n = 5$ per group except for MuRF1$^{-/-}$ KPC END mice for which $n = 4$). Data are presented as means ± SE.

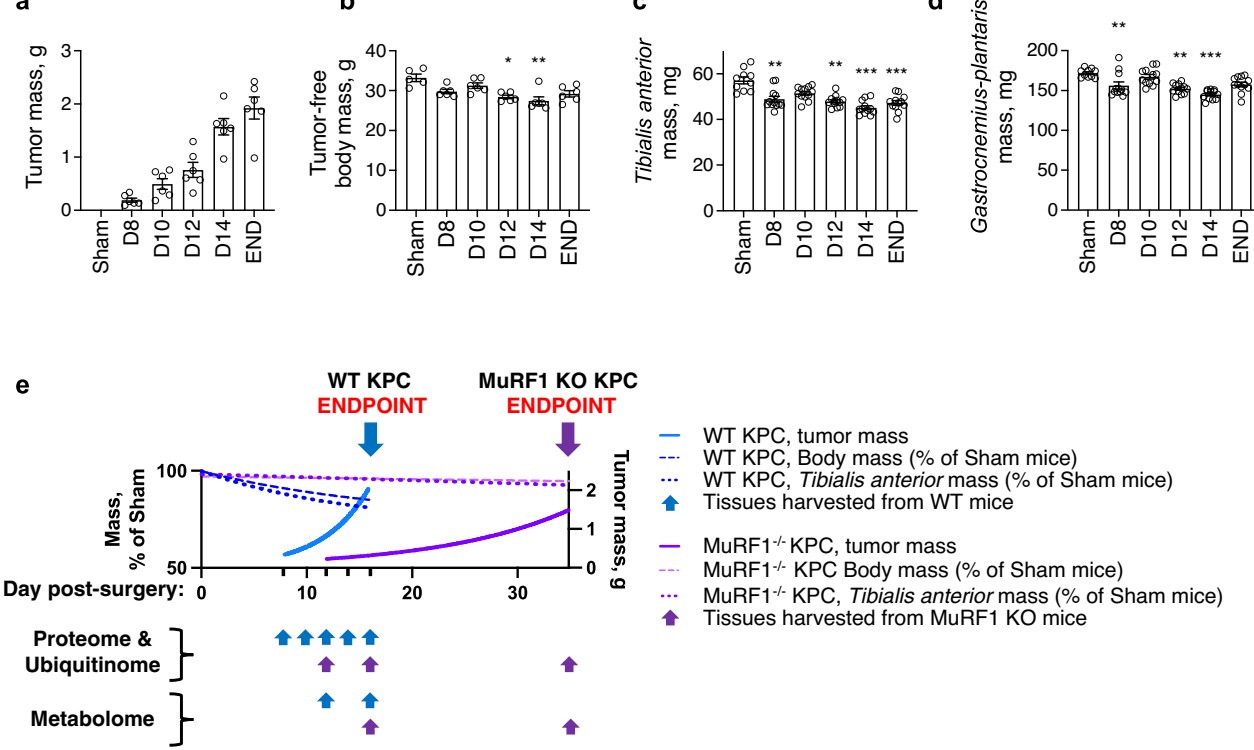

**Fig. 3 Progression of cancer cachexia in response to KPC tumors. a–d** Changes in tumor mass (**a**), tumor-free body mass (**b**, $\eta^2 = 0.4517$) and skeletal muscle mass (**c**, **d**, $\eta^2 = 0.4548$ for *tibialis anterior* and $0.4858$ for *gastrocnemius-plantaris*) in response to KPC tumor burden in WT mice. Note that data from Sham and END groups are replicated from Fig. 1 to enable comparisons. *$p < 0.05$ vs. Sham, **$p < 0.01$ vs. Sham and ***$p < 0.001$ vs. Sham. **e** Schematic illustrating the different time points at which *omics* analyses were performed. Note that for graphical representation purposes, linear and non-linear models were fitted to data from **a–d** for WT mice and from Fig. 1 for MuRF1$^{-/-}$ mice. Data are presented as means ± SE.

in MuRF1$^{-/-}$ KPC mice (Supplementary Data 2). Overall, we found that 65 to 80% of sites and 65-72% of proteins showing increased ubiquitination in response to KPC tumors were dependent upon MuRF1, either directly or indirectly (Fig. 4j). GO and KEGG pathway enrichment analyses indicated that these MuRF1 target proteins are predominantly related to cytoskeleton structure and muscle contraction (including type IIB (MYH4) and type IIX (MYH1) myosin heavy chains, as well as DES, MYPBC2, MYOM1, PLEC, TPM1, TPM2, TTN), sarcoplasmic reticulum (including ATP2A1, JPH2, RYR1, TMEM38A), glycolysis (including ALDOA, AK1, CKM, ENO3, GAPDH, LDHA, PGK1, PKM, PYGM, TPI1) and ATP binding (including AK1, CKM, INSRR, ITCH, LRRK2, MYH1, OBSCN, PKM, PYGM, TTN, UGP2, UBE2L3; Fig. 4k–o). Notably, a recent study demonstrated that MYH1 and MYH4 proteins are also ubiquitinated by another E3 ligase, UBR2, whose muscle-specific deletion prevented fast-twitch muscle wasting in response to tumor burden[29].

Lastly, we compared MuRF1 target proteins identified here to data collected by Baehr et al.[30], who evaluated changes in the skeletal muscle ubiquitinome in response to MuRF1 overexpression. This comparison identified twenty-seven sites within ten different proteins that showed increased ubiquitination in response to both KPC tumors and MuRF1 overexpression (Supplementary Fig. 2). Of these twenty-seven sites, twenty-one (78%) were ubiquitinated in response to KPC tumors through a MuRF1-dependent manner. Of these MuRF1 targets, five were related to the sarcomere (DES, MYH4, TNNT3, MYLPF and Titin), three were related to protein degradation/ubiquitination (SQSTM1, MuRF1 and VCP) and two related to glycolysis (GAPDH and LDHA).

**Reductions in proteins that support muscle protein synthesis parallel KPC-induced muscle wasting and are mediated through MuRF1.** To further identify the proteomic signature associated with the development of cancer-induced muscle wasting in WT mice, and the requirement of MuRF1 for such a signature, we next performed an unbiased TMT 10plex analysis on the *tibialis anterior*, using the same sample homogenates used for the ubiquitinome profiling (Fig. 3). Overall, we detected 58,088 peptides mapping to 1728 proteins, of which 1496 proteins were considered high confidence matches with at least two peptides mapping to them. PCA revealed good separation between genotypes and between time points (Fig. 5a). The numbers of proteins that showed an alteration in their *relative* abundance throughout tumor progression for WT and MuRF1$^{-/-}$ mice is shown in Fig. 5b.

To identify key biological pathways associated with KPC-induced muscle wasting and dysfunction, we first performed bioinformatic enrichment analysis using findings from the muscles of WT mice (Supplementary Data 3). The top 15 canonical pathways most enriched at each time point in WT mice are depicted in Supplementary Fig. 3. A positive enrichment in proteins annotating to *Acute Phase Response*, *Coagulation system*, *GP6 Signaling Pathway*, *Intrinsic Prothrombin Activation Pathway* and *LXR/RXR Activation*, as well as proteins annotating to several metabolic intermediate degradation pathways (including guanosine, adenosine, glutamate, urea, urate, inosine and 4-aminobutyrate degradation) were identified in WT KPC mice at every time point studied. Skeletal muscle from WT KPC mice further showed a negative enrichment of proteins annotating to *EIF2 signaling* and *mTOR signaling* from D12 onwards—the time point at which muscle wasting is first detectable (Fig. 3).

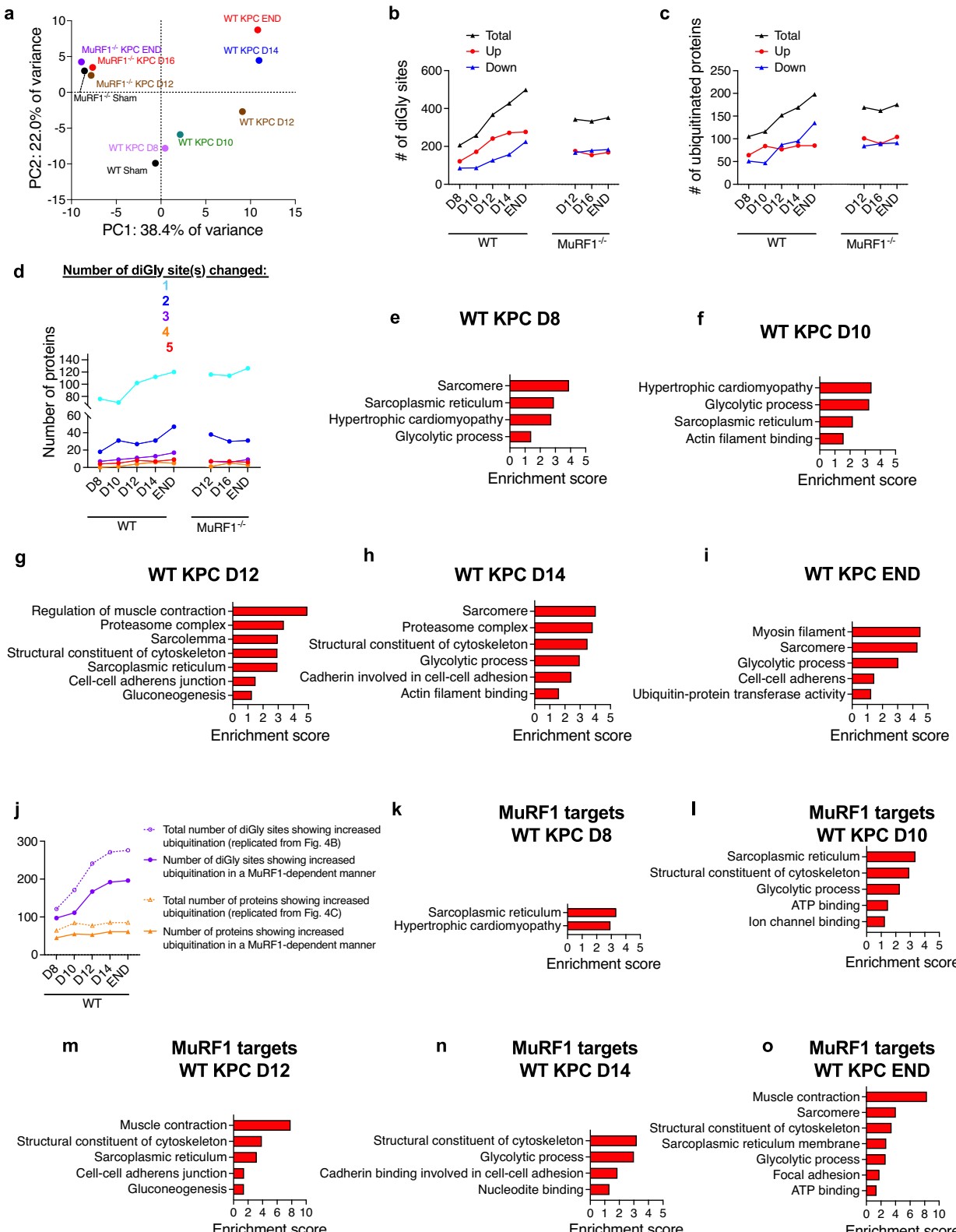

**Fig. 4 Identification of the MuRF1-dependent skeletal muscle ubiquitinome signature induced by tumor burden. a** Principal component analyses of the skeletal muscle ubiquitin-modified proteome during the progression of KPC tumor burden in WT and MuRF1$^{-/-}$ mice. **b, c** Number of diGly sites (**b**) and proteins (**c**) showing altered ubiquitination levels in each genotype in response to KPC-tumor burden. **d** Number of proteins showing altered ubiquitination levels at 1, 2, 3, 4, and ≥ 5 diGly sites in each genotype. **e–i** Enriched GO and KEGG terms associated with proteins showing increased ubiquitination in response to tumor burden in WT mice. **j** Number of diGly sites and proteins showing increased ubiquitination in response to tumor burden in skeletal muscle of WT mice but not in MuRF1$^{-/-}$ mice, indicating MuRF1-dependence. **k–o** Enriched GO and KEGG terms associated with skeletal muscle proteins whose increased ubiquitination in response to tumor burden required MuRF1. For each time point, 3-6 *tibialis anterior* muscle were pooled for analyses.

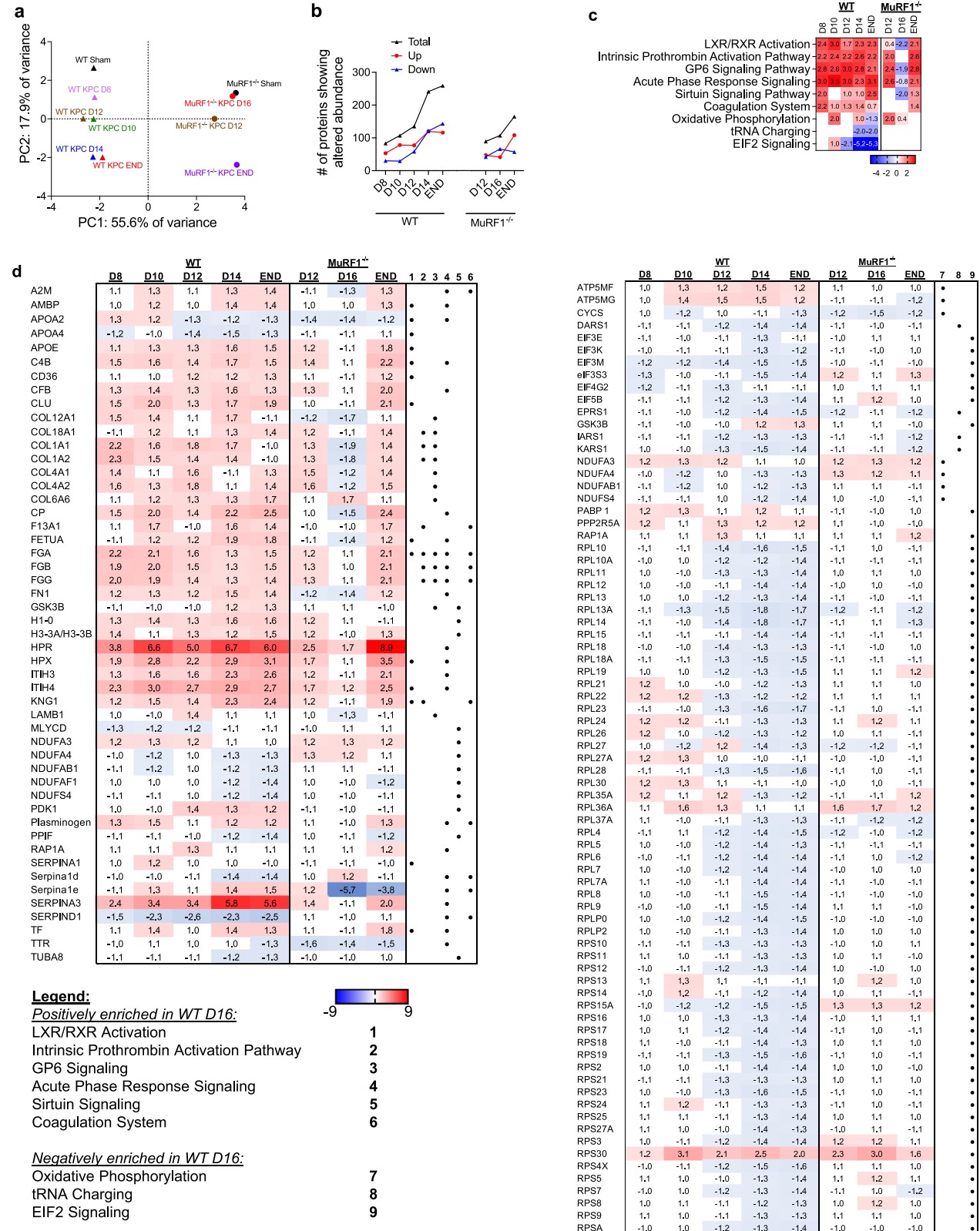

**Fig. 5 Identification of the skeletal muscle proteomic signature in response to tumor burden in the presence and absence of MURF1. a** Principal component analyses of the skeletal muscle proteome during the progression of KPC tumor burden in wild-type (WT) and MuRF1$^{-/-}$ mice. **b** Number of proteins showing altered abundance in response to KPC-tumor burden. **c** Comparison analysis of canonical pathways enriched in WT mice highlighting similarities and differences in enriched canonical pathways across the progression of cancer cachexia in WT and MuRF1$^{-/-}$ mice. **d** Abundance of proteins annotating to canonical pathways shown in **c**. For each time point, 3-6 *tibialis anterior* muscle were pooled for analyses.

A follow-up comparative analysis was conducted using Ingenuity Pathway Analysis (IPA) to identify canonical pathways that show differences in response to KPC tumors between genotypes. This analysis revealed that MuRF1 is required for the KPC-induced reduction in the relative abundance of proteins annotating to *EIF2 signaling*—including multiple ribosomal proteins—as well as *mTOR signaling, oxidative phosphorylation,* and *tRNA charging* processes. In contrast, MuRF1 deletion had minimal effects on the KPC-induced increase in proteins related to *Acute Phase Response, Coagulation system, GP6 Signaling Pathway, Intrinsic Prothrombin Activation Pathway, LXR/RXR Activation* and *Sirtuin Signaling Pathway* (Fig. 5c, d), which aligns with the expected role of MuRF1 in functioning downstream of these pathways. Overall, these analyses reveal an early and sustained upregulation of proteins involved in inflammation, immune response and extracellular matrix remodeling in response to KPC tumors that is MuRF1-independent, and a MuRF1-*dependent* downregulation of proteins involved in protein synthesis, translation initiation and oxidative phosphorylation at time points coinciding with the onset and progression of muscle wasting. Taken together, these unbiased proteomics data suggest a key link between MuRF1, an established component of the ubiquitin proteasome pathway of muscle protein degradation, and a decrease in proteins related to protein synthesis and translation initiation that associate with the initiation and progression of KPC-induced cachexia.

Of note, many of the myofibrillar and structural proteins that showed increased ubiquitination in response to KPC tumors in WT mice, whose proteolytic degradation likely contribute directly to muscle fiber atrophy, were not decreased in their *relative* abundance when compared to Sham mice (Supplementary Data 4). This was an expected finding, since normalization steps utilized for the proteomics assay included use of equal amounts of starting tissue per group, and normalization of data to *total protein signal*.

**Preventing muscle wasting via MuRF1 deletion blunts KPC-induced changes in muscle, serum and tumor metabolome.** Our finding that blocking muscle wasting via the deletion of MuRF1, which is muscle-specific, impedes KPC tumor growth suggests that muscle catabolism is required to fuel tumor growth. Based on this observation, we explored the effects of MuRF1 deletion on the tumor metabolome by conducting global metabolomic profiling *via* mass spectrometry on tumors from WT KPC mice at endpoint (day 16) and on tumors from MuRF1$^{-/-}$ on day 16 and their endpoint (day 35) (Supplementary Data 5). These analyses revealed a differential metabolic profile in tumors from cachectic mice (WT KPC) and non-cachectic mice (MuRF1$^{-/-}$ KPC) (Fig. 6a, b, Supplementary Data 6). On day 16, tumors from MuRF1$^{-/-}$ mice—which were 84% smaller than tumors from WT mice—showed a preferential accumulation of metabolites compared to tumors from WT mice. Indeed, 45 metabolites showed a higher abundance, and only 14 showed a lower abundance, in tumors from MuRF1$^{-/-}$ mice compared to tumors from WT mice (Fig. 6b, Supplementary Data 6). Alternatively stated—metabolites are more depleted in tumors from WT mice undergoing wasting than in MuRF1$^{-/-}$ mice spared from wasting. This aligns with recently published work showing that metabolites are more frequently depleted in PDAC tumors compared to adjacent non-tumor tissue[42]. Among the metabolites higher in abundance in tumors from MuRF1$^{-/-}$ mice were alpha-D-glucose, several amino acids and amino acid derivatives, and lipidic compounds (Table 1). These findings suggest that, at this time point, tumors from MuRF1$^{-/-}$ mice may be less metabolically active than tumors from WT mice. As tumors from MuRF1$^{-/-}$

mice grew and reached a size comparable to tumors from WT KPC END mice, the tendency for tumors from MuRF1$^{-/-}$ mice to accumulate metabolites remained, with 26 metabolites showing a higher abundance, compared to 11 metabolites showing a lower abundance, in tumors of MuRF1$^{-/-}$ vs. WT mice (Fig. 6b).

To obtain a better understanding of the muscle—tumor crosstalk, we also profiled the muscle (*gastrocnemii-plantaris complex*) and serum metabolome of WT and MuRF1$^{-/-}$ mice in the absence or presence of KPC tumors at time points outlined in Fig. 3 (Fig. 6c, d, Supplementary Data 7-8). Of the metabolites showing altered abundance in atrophied skeletal muscle from WT KPC mice vs. WT Sham, the predominant direction of change was a decrease (Fig. 6e, Table 2, and Supplementary Data 9). Metabolites showing reduced abundance included intermediates of the TCA cycle (malate and fumarate), amino acids, and amino acid derivatives (glycine, glutamic acid, methionine, 4-amino-butanoate, o-acetyl-l-serine, citrulline, pipecolic acid, n-acetylgly-cine, 4-hydroxyproline and 5-oxo-d-proline). In contrast, metabolites that were increased in abundance in atrophied muscles from WT KPC mice included those related to oxidative stress and redox homeostasis (methionine sulfoxide, 3-hydroxyphenyl-lactate and pyridoxamine). In comparison, in muscles from MuRF1$^{-/-}$ mice, which were protected from tumor-induced wasting, the skeletal muscle metabolome remained largely unchanged.

In serum from WT KPC mice, 36 metabolites showed an altered abundance on day 12, when cachexia is first measurable, which increased to 90 metabolites as WT KPC mice developed severe cachexia and reached endpoint (Fig. 6e, Supplementary Data 10). Among the metabolites decreased in serum in response to KPC tumors, several were related to lipid and carbohydrate metabolism (Table 3). We also observed changes in the abundance of several amino acids and amino acid derivatives, with a majority showing an increased abundance in response to KPC tumors, especially in severely cachectic mice (Table 3). Of note, amongst the 20 proteogenic amino acids, 11 showed an altered abundance in response to KPC tumors. KPC tumors also altered vitamin B and vitamin C metabolism, which are involved in antioxidative defense and energy production (Table 3). In relation to this, serum from severely cachectic mice showed higher levels of circulating 2-hydroxyphenylalanine, which is a marker of oxidative stress, and of 3-(4-hydroxyphenyl)lactate, which is an endogenous antioxidant[43] (Table 3). Similar to muscle, the serum metabolome of MuRF1$^{-/-}$ mice remained largely unchanged in response to tumor burden (Table 3).

Lastly, to obtain a better understanding of the muscle-tumor crosstalk, we integrated our metabolomic analyses from skeletal muscle, serum and tumor, and identified metabolites that showed altered abundance across tissues (Fig. 6f). The majority of these metabolites, including glycine, acyl-carnitine, serine and several carbohydrates were decreased in skeletal muscle and/or serum from cachectic WT KPC mice compared to WT Sham, suggesting a systemic depletion of these metabolites. These same metabolites were also reduced in KPC tumors from WT mice when compared to the slower-growing tumors from MuRF1$^{-/-}$ mice, suggesting that KPC tumors from WT mice may deplete these metabolites more rapidly. In support of this, glycine has been shown to be rapidly consumed by proliferating cancer cells, in order to sustain de novo purine synthesis[44]. Since purines and pyrimidines are formed simultaneously, our finding that thymidine is higher in abundance in tumors from WT mice compared to MuRF1$^{-/-}$ mice, aligns with an increase in purine synthesis. These results pave the way for future validation studies investigating the direct effects of specific MuRF1-regulated metabolites on in vivo tumor growth.

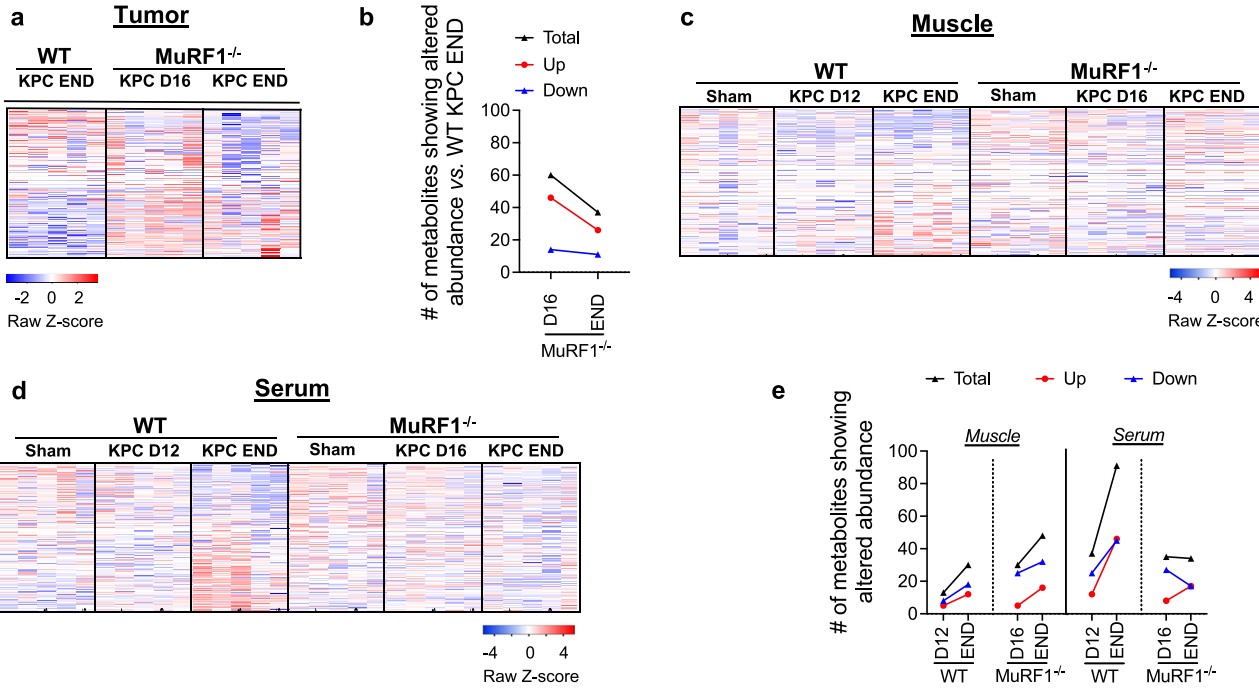

**f**

| | | Muscle | | | | Serum | | | | Tumors | |
|---|---|---|---|---|---|---|---|---|---|---|---|
| | | WT | | MuRF1⁻/⁻ | | WT | | MuRF1⁻/⁻ | | MuRF1⁻/⁻ D16 _vs._ WT END | MuRF1⁻/⁻ END _vs._ WT END |
| | | D12 | END | D16 | END | D12 | END | D16 | END | | |
| **Amino acids and derivatives** | 3-(4-HYDROXYPHENYL)LACTATE | -1.06 | 1.66 | -1.20 | -1.23 | -1.28 | 2.23 | 1.15 | -1.14 | Not Detected | |
| | 4-HYDROXY-L-PROLINE | -1.07 | -2.53 | -1.13 | -1.04 | 2.05 | 4.84 | -1.04 | -1.02 | 1.24 | 1.09 |
| | GLYCINE | -1.41 | -1.60 | -1.20 | -1.28 | -1.25 | -1.31 | -1.13 | -1.21 | 1.38 | 1.06 |
| | L-ANSERINE | 1.03 | 1.22 | 1.19 | 1.01 | -1.05 | 1.54 | 2.03 | -1.02 | -5.09 | -11.50 |
| | L-CYSTEIC ACID | 1.32 | 1.46 | 1.69 | 2.12 | Not Detected | | | | 2.55 | 1.65 |
| | L-GLUTAMIC ACID | -1.22 | -1.36 | -1.10 | -1.07 | -1.03 | -1.08 | -1.12 | -1.23 | 1.24 | -1.06 |
| | L-HISTIDINE | -1.03 | 1.02 | 1.06 | 1.09 | -1.22 | 1.22 | 1.04 | 1.03 | 1.25 | -1.05 |
| | L-HISTIDINOL | 1.03 | 1.24 | 1.08 | -1.11 | 1.48 | 2.60 | 1.23 | 1.21 | -1.28 | -1.31 |
| | L-LYSINE | 1.14 | 1.45 | -1.21 | -1.10 | -1.01 | 1.30 | -1.15 | -1.07 | 1.21 | -1.05 |
| | L-METHIONINE | -1.24 | 1.05 | -2.23 | -4.68 | 1.00 | 1.16 | -1.56 | -1.56 | -1.01 | -1.66 |
| | L-PIPECOLIC ACID | -1.30 | -1.57 | -1.10 | 1.26 | -1.15 | -1.11 | -1.10 | 1.21 | 2.29 | 2.01 |
| | L-SERINE | -1.05 | -1.05 | -1.21 | -1.18 | -1.29 | -1.17 | -1.50 | -1.57 | 1.25 | 1.19 |
| | N-ACETYLGLYCINE | -1.38 | -1.38 | 1.14 | 1.43 | -1.68 | -2.06 | 1.23 | 1.07 | 1.66 | 1.41 |
| | N-ACETYL-L-LEUCINE | 1.04 | -1.08 | 1.19 | 1.05 | -1.74 | 1.10 | 1.32 | 1.22 | 1.49 | -1.09 |
| | O-ACETYL-L-SERINE | -1.28 | -1.49 | -1.30 | -1.07 | Not Detected | | | | 1.28 | -1.10 |
| **Hexoses** | ALDO/KETO-HEXOSE | -1.08 | 1.00 | -1.19 | -1.23 | -1.03 | -1.26 | -1.08 | -1.08 | 1.31 | 1.49 |
| | ALPHA-D-GLUCOSE | -1.10 | -1.13 | -1.12 | -1.06 | -1.04 | -1.53 | -1.10 | -1.10 | 1.68 | 1.45 |
| | GLUCOSE/FRUCTOSE | -1.09 | 1.11 | -1.09 | -1.11 | -1.07 | -1.26 | -1.07 | -1.07 | 1.48 | 1.60 |
| **Others** | 2-hydroxyglutarate | -1.30 | -1.27 | -1.04 | -1.02 | -1.16 | -1.50 | -1.10 | -1.33 | 1.54 | 1.43 |
| | 4-AMINOBUTANOATE (GABA) | -1.06 | -1.57 | 1.12 | 1.04 | -1.12 | -1.26 | -1.05 | -1.14 | 1.84 | 1.34 |
| | 5'-DEOXYADENOSINE | Not Detected | | | | 1.14 | 1.70 | 1.29 | 1.21 | 2.59 | 1.21 |
| | 5-hydroxyindoleacetic acid | Not Detected | | | | 1.96 | 4.57 | 3.70 | 5.89 | 6.76 | 7.15 |
| | Acyl-Carnitine(5-OH) | -1.25 | -1.28 | -1.15 | -1.07 | -1.70 | -1.39 | -1.11 | -1.03 | 1.79 | 1.54 |
| | DEHYDROASCORBATE | -1.34 | -1.42 | -1.55 | -2.81 | -1.84 | 1.35 | 1.32 | 1.13 | 1.60 | 1.37 |
| | Glycerophosphocholine | -1.47 | -1.21 | -1.62 | -1.47 | -1.05 | -2.10 | -1.56 | -2.07 | 1.76 | 1.56 |
| | ISOCITRIC ACID | 1.06 | 1.27 | -1.13 | -1.34 | -1.19 | -1.63 | -1.10 | -1.28 | 1.99 | 1.73 |
| | PYRIDOXAMINE | 1.24 | 1.52 | -1.16 | -1.14 | 1.03 | 1.36 | -1.12 | -1.04 | Not Detected | |
| | SN-GLYCERO-3-PHOSPHOCHOLINE | -1.44 | -1.11 | -1.65 | -1.49 | Not Detected | | | | 1.72 | 1.72 |
| | THYMIDINE | 1.12 | 1.45 | 1.02 | 1.31 | 1.28 | 1.77 | 1.20 | 1.43 | -3.66 | -2.77 |
| | TRIGONELLINE | -1.46 | -1.25 | -1.09 | 1.08 | 1.28 | 1.92 | -1.34 | 1.00 | 1.47 | 1.50 |
| | URACIL | 1.24 | 1.53 | 1.11 | 1.36 | 1.28 | 3.05 | 1.46 | 1.88 | 1.17 | -1.09 |
| | XANTHOSINE | -1.11 | -1.00 | -1.13 | 1.08 | 4.71 | 4.44 | 1.21 | -7.63 | 2.01 | 1.05 |

**Fig. 6 Identification of the metabolic signatures of skeletal muscle, serum, and tumor in response to tumor burden. a** Heatmap showing the relative abundance of all known metabolites detected in tumors of WT and MuRF1⁻/⁻ mice. **b** Number of metabolites presenting altered abundance ($p < 0.05$ and FC > |1.2|) in tumors of WT and MuRF1⁻/⁻ mice. **c, d** Heatmaps showing the relative abundance of all known metabolites detected in skeletal muscle (_gastrocnemius-plantaris_, **c**) and serum (**d**) of WT and MuRF1⁻/⁻ mice. **e** Number of metabolites presenting altered abundance ($p < 0.05$ and FC > |1.2|) in skeletal muscle (_gastrocnemius-plantaris_) and serum of WT and MuRF1⁻/⁻ mice. **f** Metabolites showing altered abundance in response to tumor burden in at least two tissues in the WT mice. In **a**, **c**, **d**, and **f**, red indicates increased abundance and blue indicates reduced abundance compared to the averaged abundance of the respective Sham groups. In **f**, the average value is presented when metabolites were detected in both negative and positive phases. For each time point, 5 biological replicates were analyzed.

**Table 1 Metabolites showing differential abundance in tumors from MuRF1$^{-/-}$ vs. WT mice.**

| | MuRF1$^{-/-}$ D16 vs. WT END | MuRF1$^{-/-}$ END vs. WT END |
|---|---|---|
| L-anserine | -5.09 | **-11.50** |
| Thymidine | **-3.66** | **-2.77** |
| Hypoxanthine | **-1.74** | **-2.50** |
| Glu-thr | 1.06 | **-1.94** |
| L-tryptophan | **-1.32** | **-1.73** |
| Tryptophan | **-1.32** | **-1.71** |
| L-tryptophan-nh3 | **-1.32** | **-1.70** |
| L-methionine | -1.01 | **-1.66** |
| 2-hydroxybutyric acid | **-1.57** | **-1.63** |
| Putrescine | **-1.53** | **-1.62** |
| Alpha-aminoadipate/n-methyl-l-glutamate | **-1.37** | **-1.61** |
| O-phospho-l-serine | **-1.38** | **-1.55** |
| Dihydroxyacetone phosphate | **-1.62** | **-1.44** |
| Alanine/sarcosine | -1.08 | **-1.41** |
| Stachydrine | **1.69** | -1.19 |
| Phosphocholine | **-1.36** | -1.17 |
| Choline | **-1.23** | -1.13 |
| O-acetyl-l-serine | **1.28** | -1.10 |
| N-acetyl-l-leucine | **1.49** | -1.09 |
| L-glutamic acid | **1.24** | -1.06 |
| L-histidine | **1.25** | -1.05 |
| Sn-glycerol 3-phosphate | **-1.62** | -1.05 |
| Glycerol 2-phosphate/sn-glycerol 3-phosphate | **-1.49** | -1.05 |
| Aldopentose | **1.30** | 1.00 |
| Xanthosine | **2.01** | 1.05 |
| Glycine | **1.38** | 1.06 |
| Cytosine | **2.30** | 1.09 |
| L-isoleucine | **1.26** | 1.10 |
| Leucine | **1.33** | 1.13 |
| Isocytosine | **2.45** | 1.13 |
| Cortisol 21-acetate | **1.34** | 1.17 |
| Dl-5-hydroxylysine | **1.32** | 1.18 |
| L-serine | **1.25** | 1.19 |
| 5'-deoxyadenosine | **2.59** | 1.21 |
| Propanoylcarnitine | **1.45** | 1.25 |
| Glyceraldehyde | 1.24 | **1.34** |
| D-saccharic acid | **1.42** | **1.34** |
| 4-aminobutanoate (gaba) | **1.84** | **1.34** |
| Hypotaurine | **2.53** | 1.35 |
| Dehydroascorbate | **1.60** | **1.37** |
| 2-aminophenol | **1.40** | **1.41** |
| N-acetylglycine | **1.66** | **1.41** |
| 2-hydroxyglutarate | **1.54** | **1.43** |
| Alpha-d-glucose | **1.68** | **1.45** |
| Aldo/keto-hexose | 1.31 | **1.49** |
| Trigonelline | **1.47** | 1.50 |
| 2-amino-2-methylpropanoate | **2.14** | 1.53 |
| Acyl-carnitine(5-oh) | **1.79** | **1.54** |
| 3-aminoisobutanoate | **2.10** | **1.54** |
| Glycerophosphocholine | **1.76** | **1.56** |
| L-cysteine | **2.84** | 1.57 |
| 12(s)-hete | **2.32** | 1.59 |
| Glucose/fructose | **1.48** | **1.60** |
| L-cysteic acid | **2.55** | 1.65 |
| Butanoylcarnitine | **1.69** | 1.67 |
| Thromboxane B2 | **1.98** | **1.68** |
| Sn-glycero-3-phosphocholine | **1.72** | **1.72** |
| Isocitric acid | **1.99** | **1.73** |
| Ascorbic acid-2h | **1.69** | **1.74** |
| Malonate | **1.51** | **1.76** |
| Sulfoacetaldehyde | **1.94** | **1.91** |
| L-pipecolic acid | **2.29** | **2.01** |
| Adenine | **-1.65** | 2.39 |
| Guanidinoacetate | **3.91** | **2.59** |
| 2-methylbutyroylcarnitine | 2.11 | **3.28** |
| 2-methylmaleate | **10.80** | **3.93** |
| 5-hydroxyindoleacetic acid | **6.76** | **7.15** |

Bold values indicate the change was statistically significant. Blue (for metabolites showing a lower abundance) and red (for metabolites showing higher abundance) color gradients were applied to highlight the magnitude of fold changes.

**Table 2 Metabolites showing altered abundance in response to tumor burden in skeletal muscle of WT mice.**

| | WT | | MuRF1-/- | |
|---|---|---|---|---|
| | D12 | END | D16 | END |
| 4-hydroxy-l-proline | -1.07 | **-2.53** | -1.13 | -1.04 |
| 4-hydroxyproline | -1.02 | **-2.24** | 1.12 | 1.18 |
| Glycine | **-1.41** | **-1.60** | -1.20 | **-1.28** |
| L-pipecolic acid | **-1.30** | **-1.57** | -1.10 | 1.26 |
| 4-aminobutanoate (gaba) | -1.06 | **-1.57** | 1.12 | 1.04 |
| O-acetyl-l-serine | **-1.28** | **-1.49** | **-1.30** | -1.07 |
| (R)-malate | -1.14 | **-1.47** | -1.13 | -1.04 |
| Citrulline | -1.07 | **-1.47** | -1.14 | 1.04 |
| N-acetylglycine | **-1.38** | **-1.38** | 1.14 | **1.43** |
| L-glutamic acid | -1.22 | **-1.36** | -1.10 | -1.07 |
| Fumarate | -1.11 | **-1.36** | -1.14 | -1.01 |
| Malate | -1.11 | **-1.34** | **-1.36** | -1.19 |
| 2-hydroxyglutarate | **-1.30** | **-1.27** | -1.04 | -1.02 |
| Uridine | -1.14 | **-1.25** | -1.02 | 1.02 |
| Glycerophosphocholine | **-1.47** | -1.21 | **-1.62** | **-1.47** |
| 5-oxo-d-proline | **-1.25** | -1.19 | -1.19 | -1.19 |
| Sn-glycero-3-phosphocholine | **-1.44** | -1.11 | **-1.65** | **-1.49** |
| L-methionine | **-1.24** | 1.05 | **-2.23** | **-4.68** |
| Methionine sulfoxide | **1.24** | **1.21** | 1.12 | **1.33** |
| L-anserine | 1.03 | **1.22** | 1.19 | 1.01 |
| L-histidinol | 1.03 | **1.24** | 1.08 | -1.11 |
| D-(+)-cellobiose | -1.10 | **1.33** | -1.34 | **-2.57** |
| L-lysine | 1.14 | **1.45** | -1.21 | -1.10 |
| Thymidine | 1.12 | **1.45** | 1.02 | **1.31** |
| L-cysteic acid | **1.32** | **1.46** | **1.69** | **2.12** |
| Pyridoxamine | **1.24** | **1.52** | -1.16 | -1.14 |
| Uracil | **1.24** | **1.53** | 1.11 | **1.36** |
| 3-(4-hydroxyphenyl)lactate | -1.06 | **1.66** | -1.20 | **-1.23** |
| L-kynurenine | **1.31** | **2.54** | 1.14 | **1.72** |

For WT tumor-bearing groups, values are fold-change compared to WT Sham, while for MuRF1-/- tumor-bearing groups values are fold-change compared to MuRF1-/- Sham. Bold values indicate the change was statistically significant. Blue (for metabolites showing a lower abundance) and red (for metabolites showing higher abundance) color gradients were applied to highlight the magnitude of fold changes.

**MuRF1 deletion also protects against KPC-induced wasting, slows tumor growth and extends survival in female mice.** To determine whether the protection conferred by MuRF1 deletion against cancer cachexia in male mice is similarly conferred in female mice, we conducted a final experiment in which female WT and MuRF1⁻/⁻ mice were inoculated with KPC cells and tracked until each mouse reached its humane endpoint (Fig. 7). Importantly, these experiments were conducted in a different lab (Doles Lab at Mayo Clinic) to those in male mice (Judge Lab at University of Florida) and utilized a different PDAC cell line (KPC T42D vs. KPC 1245). Thus, this experiment not only considers sex as a biological variable, but also accounts for geographical location/lab specificity and KPC cell line used. Our findings show that, similar to male mice, MuRF1 deletion in female mice extended maximum survival—to 64 days post inoculation, from 39 days in WT mice (Fig. 7a), and slowed tumor growth (Fig. 7b). Further, as tumors grew, both lean mass and fat mass were greater in MuRF1⁻/⁻ vs. WT mice (Fig. 7c, d), and tissue mass of the *tibialis anterior*, *gastrocnemius* and hearts were all significantly greater in MuRF1⁻/⁻ mice compared to WT mice at their respective endpoints (Fig. 7e–g). Moreover, muscle fiber cross-sectional area was significantly larger in *tibialis anterior* muscle from MuRF1⁻/⁻ KPC compared to those from WT KPC mice (Fig. 7h, i). Although *omics* analyses were not similarly conducted as in male mice, these findings in female mice, using a different KPC cell line and conducted in a different lab, lend strong support for the role of MuRF1 in mediating cancer-associated muscle wasting and tumor growth.

## Conclusions
Overall, our findings demonstrate that, in the context of KPC tumors, loss of the muscle-specific E3 ubiquitin ligase, MuRF1,

protects against skeletal muscle and fat wasting, prevents skeletal muscle dysfunction, slows the rate of tumor growth, and extends survival. Our unbiased *omics* analyses in skeletal muscle reveal that the protection afforded through loss of MuRF1 are related to not only the role of MuRF1 in mediating muscle protein ubiquitination and degradation but also, to the depression of pathways that support protein synthesis and disruptions to muscle metabolism. Through these muscle-specific functions, our data suggest that MuRF1 deletion may slow tumor growth through depriving the tumor of key energy substrates necessary for rapid tumor growth. Importantly, to our knowledge, these findings are the first to establish, albeit in a mouse model, that directly interfering with skeletal muscle wasting can alter the tumor metabolome, slow tumor growth and extend survival.

## Methods
**Animals.** Male and female C57BL/6J wild-type mice (hereafter referred as WT) were purchased from Jackson laboratory (Bar Harbor, Maine, USA). MuRF1 knockout (Trim63^tm1Glas–RRID: MGI3785746; MuRF1⁻/⁻) have previously been described in ref. [20]. All experiments using male mice were conducted at the University of Florida in 8- to 15-week-old mice derived from a colony generously provided by Regeneron Pharmaceuticals[20]. All experiments using female mice were conducted at the Mayo Clinic in 7- to 8-week-old mice kindly provided by Dr. Sue Bodine (University of Iowa). All mouse studies were performed in compliance with the National Institutes of Health Guidelines for Use and Care of Laboratory Animals and approved by the University of Florida and Mayo Clinic Institutional Animal Care and Use. Mice were housed in a temperature-controlled and humidity-controlled facility on a 12 h light/dark cycle with *ad libitum* access to water and standard diet.

**Cancer cell culture and inoculation.** Pancreatic cancer cell lines were a kind gift of Dr. David Tuveson (Cold Spring Harbor Laboratory, Cold Spring Harbor, New York, NY). KPC FC1245 and KPC T42D cancer cells (hereafter referred to as KPC) were derived from a LSL-Kras^G12D/+; LSL-Trp53^R172H/+; Pdx-1-Cre mouse backcrossed to the C57BL/6 genetic background[45,46]. KPC cells were cultured in

**Table 3 Metabolites showing altered abundance in response to tumor burden in serum of WT mice.**

| | WT | | MuRF1-/- | |
|---|---|---|---|---|
| | D12 | END | D16 | END |
| 2'-deoxyguanosine | -1.64 | -5.44 | 1.55 | -1.25 |
| Sorbate | -2.28 | -4.55 | -1.03 | -1.09 |
| Eicosapentaenoic acid | -1.16 | -3.01 | -1.29 | -1.58 |
| Lysope (20:4) | -1.87 | -2.18 | -1.16 | -1.42 |
| Arachidonic acid (20:4) | -1.15 | -2.16 | -1.25 | -1.58 |
| Lysope (22:6) | -1.81 | -2.11 | -1.23 | -1.50 |
| Glycerophosphocholine | -1.05 | -2.10 | -1.56 | -2.07 |
| Lysopc (20:4) | -1.67 | -2.08 | -1.12 | -1.15 |
| N-acetylglycine | -1.68 | -2.06 | 1.23 | 1.07 |
| 2-methylglutaric acid | 1.00 | -2.01 | -1.08 | -1.42 |
| 6-carboxyhexanoate | 1.14 | -1.91 | -1.17 | -1.42 |
| Docosahexaenoic acid (22:6) | -1.32 | -1.91 | -1.30 | -1.64 |
| Linoleic acid (18:2) | -1.33 | -1.84 | -1.21 | -1.56 |
| Mono-(2-ethylhexyl) phthalate dimer | -1.05 | -1.79 | -1.23 | -1.30 |
| Phenyl acetate | 1.03 | -1.75 | -1.12 | -1.50 |
| Suberic acid | 1.12 | -1.70 | -1.03 | -1.33 |
| 2-deoxy-D-galactose (fructose/glucose) | -1.64 | -1.65 | -1.55 | -1.26 |
| Isocitric acid | -1.19 | -1.63 | -1.10 | -1.28 |
| 3-tert-Butyladipic acid | 1.01 | -1.61 | -1.00 | -1.20 |
| Alpha-d-glucose | -1.04 | -1.53 | -1.10 | -1.10 |
| Indole-3-acetamide | 1.07 | -1.52 | -1.09 | -1.27 |
| L-lactic acid | -1.21 | -1.51 | -1.17 | -1.12 |
| 2-hydroxyglutarate | -1.16 | -1.50 | -1.10 | -1.33 |
| Sulcatol | 1.01 | -1.50 | -1.18 | -1.20 |
| Lysopc (16:0) | -1.47 | -1.48 | -1.12 | -1.15 |
| 2,5-dihydroxybenzoate | -1.14 | -1.42 | -1.30 | -1.11 |
| Acyl-carnitine(5-oh) | -1.70 | -1.39 | -1.11 | -1.03 |
| Azelaic acid | 1.07 | -1.38 | 1.00 | -1.23 |
| 5-Hydroxymethyl-2-furaldehyde | -1.09 | -1.36 | -1.09 | -1.12 |
| Ascorbate | -1.13 | -1.33 | -1.13 | -1.13 |
| Betaine | -1.04 | -1.33 | -1.13 | -1.14 |
| D-mannosamine | -1.04 | -1.33 | 1.01 | -1.05 |
| Pyridoxine | -1.12 | -1.32 | 1.20 | 1.02 |
| Glycine | -1.25 | -1.31 | -1.13 | -1.21 |
| D-(+)-galacturonic acid | -1.13 | -1.31 | -1.09 | -1.05 |
| 4-aminobutanoate (gaba) | -1.12 | -1.26 | -1.05 | -1.14 |
| Glucose/fructose | -1.07 | -1.26 | -1.07 | -1.07 |
| Aldo/keto-hexose | -1.03 | -1.26 | -1.08 | -1.08 |
| D-glucuronic acid | -1.09 | -1.24 | -1.09 | -1.08 |
| Nicotinamide | -1.63 | -1.24 | 1.09 | 1.07 |
| 3-hydroxy-3-methylglutarate | 1.16 | -1.24 | 1.06 | -1.06 |
| Arabinose | -1.03 | -1.22 | -1.04 | -1.14 |
| L-serine | -1.29 | -1.17 | -1.50 | -1.57 |
| 4-hydroxyphenylacetate | -1.53 | -1.09 | -1.32 | -1.18 |
| 3-Hydroxydecanoic acid | -1.80 | -1.05 | 1.11 | -1.81 |
| L-asparagine | -1.45 | -1.04 | -1.53 | -1.62 |
| Asparagine | -1.60 | 1.02 | -1.52 | -1.68 |
| 5-hydroxyindoleacetate | -1.38 | 1.02 | 1.17 | -1.39 |
| Carnosine | -1.28 | 1.02 | 1.15 | -1.11 |
| Lysope (18:0) | -1.54 | 1.03 | -1.10 | 1.06 |
| Urocanate | -1.76 | 1.05 | -1.38 | -1.46 |
| Urate | 1.35 | 1.06 | 1.05 | -1.57 |
| N-acetyl-l-leucine | -1.74 | 1.10 | 1.32 | 1.22 |
| L-arginine | 1.53 | 1.10 | 1.53 | 1.46 |
| L-histidine | -1.22 | 1.22 | 1.04 | 1.03 |
| L-lysine | -1.01 | 1.30 | -1.15 | -1.07 |
| 3-hydroxyanthranilic acid | -1.04 | 1.32 | -1.08 | 1.65 |
| Beta-alanine | -1.20 | 1.33 | -1.16 | -1.22 |
| 3-sulfino-l-alanine | -1.39 | 1.33 | 1.23 | -1.31 |
| Dehydroascorbate | -1.84 | 1.35 | 1.32 | 1.13 |
| Pyridoxamine | 1.03 | 1.36 | -1.12 | -1.04 |
| Tiglylcarnitine | -1.44 | 1.41 | -1.11 | -1.13 |
| Lumichrome | -1.20 | 1.43 | -1.01 | 1.08 |
| 2-hydroxyphenylacetic acid | -1.87 | 1.44 | 1.13 | -1.30 |
| Allantoin | 1.07 | 1.45 | 1.19 | 1.31 |
| L-proline | -1.03 | 1.48 | -1.43 | -1.48 |
| Citramalate | 1.44 | 1.51 | -1.03 | 1.18 |
| 2-hydroxyphenylalanine | -1.04 | 1.52 | -1.24 | -1.39 |
| Alpha-aminoadipate | -1.08 | 1.53 | -1.03 | -1.01 |
| Taurine | -1.07 | 1.53 | 1.23 | -1.24 |
| N-acetyl-dl-serine | -1.12 | 1.55 | 1.09 | 1.06 |
| Glycyl-L-leucine | 1.48 | 1.58 | 1.05 | -1.15 |
| D-glucosamine 6-phosphate | 1.07 | 1.61 | 1.26 | 1.14 |
| Proline | -1.10 | 1.63 | -1.46 | -1.64 |
| L-tyrosine | -1.04 | 1.65 | -1.27 | -1.49 |
| 5'-deoxyadenosine | 1.14 | 1.70 | 1.29 | 1.21 |
| Spermidine | 1.47 | 1.73 | -1.44 | -1.15 |
| 2-hydroxy-4-(methylthio)butyric acid | -1.04 | 1.73 | 1.31 | -1.05 |
| Pantothenic acid | 1.22 | 1.74 | -1.27 | -1.11 |
| 4-imidazoleacetic acid | 1.01 | 1.76 | -1.13 | 1.00 |
| Thymidine | 1.28 | 1.77 | 1.20 | 1.43 |
| D-pantothenic acid | 1.28 | 1.79 | -1.23 | -1.09 |
| N-acetyl-l-alanine | -1.01 | 1.90 | 1.19 | 1.17 |
| Trigonelline | 1.28 | 1.92 | -1.34 | 1.00 |
| Orotate | 1.31 | 1.95 | 1.20 | -1.34 |
| Methyl beta-d-galactoside | 1.24 | 1.96 | -1.02 | 1.02 |
| 4-acetamidobutanoate | 1.15 | 2.06 | -1.13 | 1.16 |
| N-methyl-l-histidine | 1.27 | 2.08 | 1.01 | 1.17 |
| 4-hydroxybenzaldehyde | 1.57 | 2.08 | -1.44 | -1.08 |
| Hippurate | 1.58 | 2.10 | -1.44 | -1.08 |
| 4-guanidinobutanoate | 1.05 | 2.10 | -1.74 | 1.04 |
| Phenylacetic acid | -1.37 | 2.18 | 1.17 | -1.19 |
| 3-(4-hydroxyphenyl)lactate | -1.28 | 2.23 | 1.15 | -1.14 |
| Formyl-l-methionyl peptide | 1.25 | 2.25 | 1.22 | 1.19 |
| L-histidinol | 1.48 | 2.60 | 1.23 | 1.21 |
| Leu pro | 3.02 | 3.00 | 1.22 | 1.08 |
| Ferulate | 1.56 | 3.01 | -3.32 | -3.82 |
| Uracil | 1.28 | 3.05 | 1.46 | 1.88 |
| Thymine | 1.31 | 3.11 | -1.10 | 1.22 |
| 1-methylnicotinamide | -1.20 | 3.20 | 1.69 | 1.99 |
| N-acetyl-l-cysteine | 1.17 | 4.13 | -1.77 | -1.38 |
| 4-methylcatechol | 1.55 | 4.28 | -1.08 | -1.37 |
| Xanthosine | 4.44 | 4.44 | 1.21 | -7.63 |
| 5-hydroxyindoleacetic acid | 1.96 | 4.57 | 3.70 | 5.89 |
| 4-hydroxy-l-proline | 2.05 | 4.84 | -1.04 | -1.02 |
| L-arabitol | -1.19 | 24.61 | 3.17 | 3.31 |

For WT tumor-bearing groups, values are fold-change compared to WT Sham, while for MuRF1-/- tumor-bearing groups values are fold-change compared to MuRF1-/- Sham. Bold values indicate the change was statistically significant. Blue (for metabolites showing a lower abundance) and red (for metabolites showing higher abundance) color gradients were applied to highlight the magnitude of fold changes.

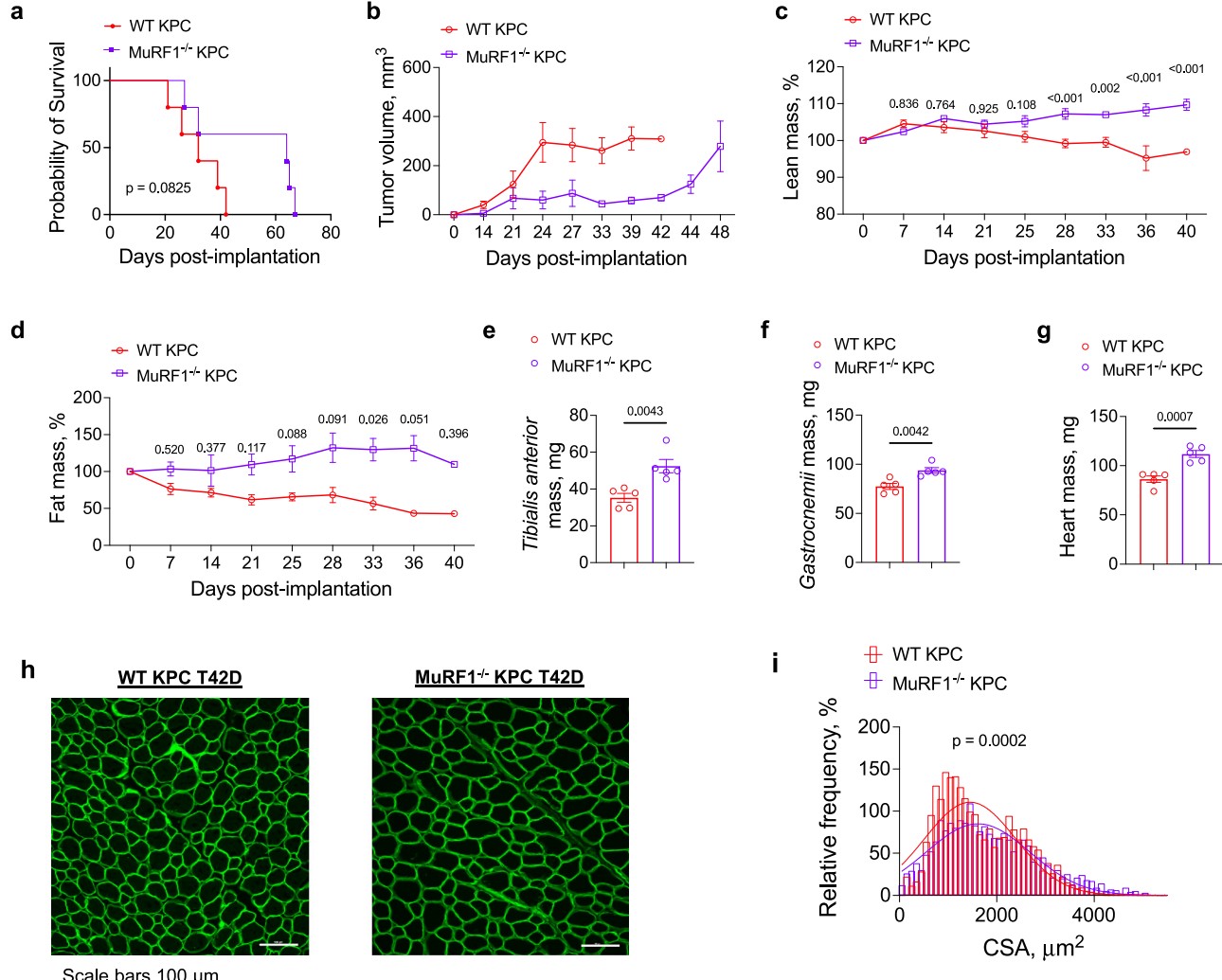

**Fig. 7 MuRF1 deletion also protects against KPC T42D-induced wasting, slows tumor growth, and extends survival in female mice. a, b** MuRF1 deletion extends maximal survival in response to KPC T42D tumors (**a**) and slows tumor growth (**b**). **c, d** Lean (**c**) and fat (**d**) mass was measured *via* Echo MRI throughout tumor progression until the last WT mice reached its endpoint. **e, f** MuRF1$^{-/-}$ KPC T42D mice present heavier *tibialis anterior* (**e**, $\eta^2 = 0.6601$), *gastrocnemius* (**f**, $\eta^2 = 0.6609$) and heart (**g**, $\eta^2 = 0.7778$) compared to WT KPC T42D mice. **h, i** Representative images of *tibialis anterior* cross-sections stained with laminin (**h**) revealing larger fibers (**i**) in MuRF1$^{-/-}$ vs. WT KPC T42D mice ($R^2 = 0.8242$ in WT and 0.9407 in MuRF1$^{-/-}$ mice). In **i**, cross-sectional area data were binned, fit with a Gaussian least squares regression and significance was determined by calculating the extra sum-of-squares *F*-test. Data are presented as means ± SE. $N = 5$ per group.

Dulbecco's Modified Eagle Medium supplemented with 10% fetal bovine serum, 1% penicillin and 1% streptomycin at 37 °C in a 5% $CO_2$ humidified chamber.

Eight to 15 weeks old male WT and MuRF1$^{-/-}$ mice were orthotopically injected with $2.5 \times 10^5$ KPC FC1245 cells diluted into 50 µl of sterile phosphate buffer saline (PBS) solution (KPC groups) or PBS alone (Sham groups) as described in ref. [39]. Seven to 8 weeks old female WT and MuRF1$^{-/-}$ mice were orthotopically inoculated with $0.5 \times 10^4$ KPC T42D cells. Male mice were euthanized at different stages of tumor progression, as outlined in the results, while all female mice were euthanized at their respective Institutional Animal Care and Use Committee (IACUC)-mandated humane endpoint determined based on body condition score, body mass loss and signs of pain and/or distress.

**Body composition measurements.** Longitudinal body composition including lean mass and fat mass was measured *via* EchoMRI Body Composition Analyzer (Echo Medical Systems) as previously described[47]. Longitudinal tumor measurements were performed *via* caliper measurements.

**Ex vivo skeletal muscle function assessment.** Ex vivo skeletal muscle contractile properties were assessed as previously described[48] at the Physiological Assessment Core of the University of Florida. Briefly, freshly isolated *soleus* muscles and diaphragm strips were mounted on a force transducer (dual-mode lever system—Aurora Scientific, Ontario, Canada) placed in a 22 °C bath of Ringers solution equilibrated with 95% $O_2$ and 5% $CO_2$. After determination of muscle optimal

length, maximum isometric twitch and tetanic forces were determined using a single supramaximal stimulation and a 500-ms stimulation train at 150 Hz, respectively. Set of stimulations were separated by a 5-min rest period. Force-frequency relationships were then built by stimulating *soleus* muscles and diaphragm strips at 5, 10, 20, 30, 50, 80, 120, and 150 Hz. Subsequently, resistance to fatigue was assessed by stimulating *soleus* muscles once per second for 10 min (200-µs pulse width, 100 Hz, 330-ms duration).

**Immunohistochemistry**

*Male experiments.* Immediately upon harvest, *tibialis anterior* and *soleus* muscles were embedded in optimal cutting temperature compound and frozen in liquid isopentane cooled in liquid nitrogen before being stored at –80 °C. Skeletal muscle were cut across the midbelly and sectioned at –20 °C using a Microm HM 550 Cryostat (Microm International, Walldorf, Germany) to obtain cross-sections of 10-µm thickness that were stored at –80 °C until being stained. Prior to staining, cross-sectioned were thawed and air-dried for 30 min. *Tibialis anterior* cross-sections were then rehydrated for 5 min in PBS, blocked for 30 min in 25% SuperBlock blocking buffer diluted in PBS (ThermoFisher, Waltham, Massachusetts, USA), incubated for 1 h at room temperature with wheat-germ agglutinin (WGA) conjugated with Alexa Fluor 594$^{TM}$ (ThermoFisher) diluted 1:300 in PBS and washed 3 × 5 min with PBS as previously described in ref. [48]. *Soleus* cross-sections were rehydrated for 5 min in PBS, blocked for 30 min in 25% SuperBlock blocking buffer diluted in PBS (ThermoFisher), incubated for 1h30 at room

temperature with antibodies against myosin heavy chain I (BA-D5, Developmental Studies Hybridoma Bank (DHSB), Iowa City, Iowa, USA) and myosin heavy chain IIA (SC-71, DHSB) diluted 1:10 in SuperBlock buffer solution diluted 1:20 in PBS, washed 3 × 5 min with PBS, incubated for 1 h at room temperature with appropriate secondary antibodies and WGA conjugated with Alexa Fluor 594[TM] diluted 1:300 in PBS, and washed 3 × 5 min with PBS[48]. All sections were imaged with a Leica TCS SP8 microscope (Leica Microsystems, Weltzar, Germany) using a 20x objective. Images were analyzed with ImageJ (National Institutes of Health, Bethesda, MD, USA). Skeletal muscle fiber cross-sectional area was measured using a semi-automated threshold analysis based on the WGA signal to delineate the borders of individual fibers as previously described[48]. A total of 2186–3190 (mean: 2514) were measured for each *tibialis anterior* section and 343–976 (mean: 730) for each *soleus* section.

*Female experiments*. *Tibialis anterior* harvested from female mice were placed in a sucrose sink (30%) overnight prior to freezing, sectioning and laminin staining as previously described[47]. Briefly, sections (8 μm) were fixed in 4% paraformaldehyde (PFA) for 5 min at room temperature prior to immunostaining. Once fixed, tissues were stained with rat anti-laminin (MilliporeSigma, 4HB-2). Next, the Alexa Fluor 488 (Invitrogen) conjugates were used as secondary antibodies. Three non-overlapping fields per tissue were imaged at 20x. Myovision software was utilized to measure *tibialis anterior* minimum feret diameters as previously described[49].

**Blood collection**. Blood was collected from the aortic artery, mixed with clotting factors (Microtainer BD 365967, BD, Franklin Lakes, New Jersey, USA), incubated at room temperature for at least 30 min and centrifuged at 2500 × g for 10 min at 4 °C. Collected serum samples were then stored at –80 °C until analysis.

**Proteomics**

*Sample preparation*. Freshly harvested *tibialis anterior* muscles were quickly rinsed in PBS and snaped-frozen in liquid nitrogen before being stored at –80 °C until analysis. For each group, approximately equal amount of starting tissue ( ~ 200 mg, $N = 3–6$ samples/group) were pooled and sent to Cell Signaling Technologies, Danvers, Massachusetts, USA) for protein extraction, and subsequent global proteome and ubiquitinome analyses according to standard procedures[50,51].

*Global proteome profiling*. The methodological procedures for global proteomic quantification were described in details in ref. [50]. Briefly, pooled samples were labeled with Tandem Mass Tag (TMT) reagents (Thermo Fisher Scientific, Waltman, Massachusetts, USA), combined and fractionated using basic reverse phase (bRP) fractionation chromatography. 96 bRP fractions were collected over the entire gradient and concatenated into 24 fractions that were run on an Orbitrap-Fusion Lumos Tribridmass spectrometer (Thermo Fisher Scientific) for LC-MS/MS analysis with Multi-Notch MS3 Quantification. MS/MS spectra were evaluated using Comet[52] and the core platform from Harvard University. Searches were performed against the most recent update of the Uniprot *Mus musculus* database with mass accuracy of ± 50 ppm for precursor ions and 0.02 Da for product ions. Results were filtered with mass accuracy of ± 5 ppm on precursor ions and presence of intended motif. Protein identification was further subjected to a 1% false discovery rate. For WT and MuRF1[-/-] mice respectively, protein abundance was evaluated by computing fold-change (FC) between tumor-bearing groups vs. Sham. Proteins were considered as differentially abundant if they met the following criteria: detected peptides ≥ 2 and absolute FC > 1.25. Proteins showing an altered relative abundance were subsequently submitted to bioinformatic analyses to identify enriched pathways using Ingenuity Pathway Analysis software[53]. Principal component analyses were conducted using the R built-in prcomp function on the summed signal to noise ratio (R version 4.0.3[54]).

*Global ubiquitinome profiling*. The methodological procedures used for global ubiquitinome analyses are described in details in ref. [51], and were previously used to analyze skeletal muscle ubiquitinome by us[41] and others[30]. Briefly, extracted proteins were digested with trypsin to generate peptides and di-glycine remnants (diGly), reversed phase purified, enriched with the PTMScanR Ubiquitin Remnant Motif (K-ε-GG) Kit (Cell Signaling Technology) and run in LC-MS/MS on an Orbitrap-Fusion Lumos Tribridmass spectrometer (Thermo Fisher Scientific). The same methods as described above for the global proteome profiling were used to assign spectra to peptides and proteins. Only diGly modifications annotating to unique proteins were retained for further analyses. For these modifications, FC were computed between WT tumor-bearing vs. WT Sham mice and MuRF1[-/-] tumor-bearing vs. MuRF1[-/-] Sham mice. Differences in ubiquitination-like modifications were deemed significant if the following criteria were met: absolute FC > 2 for modified diGly sites presenting a signal intensity > 1,000,000 or absolute FC > 2.5 for modified diGly sites presenting a signal intensity > 500,000. Principal component analyses were conducted using the R built-in prcomp function on normalized signal (R version 4.0.3[54]). Bioinformatic enrichment analyses were performed with DAVID[55,56] using GO[57,58] and KEGG[59] annotations.

All data have been deposited to the ProteomeXchange Consortium[60,61] *via* the PRIDE partner repository[62]. The global proteome and ubiquitinome respective datasets are accessible with the identifiers PXD039158 and PXD039151.

**Metabolomics**. Freshly harvested *gastrocnemius-plantaris* muscle complex and tumor samples were quickly rinsed in PBS, snaped-frozen in liquid nitrogen and stored at –80 °C until analysis. Five samples per group and per tissue were sent to the Southeast Center for Integrated Metabolomics of the University of Florida. Skeletal muscle and tumor samples were extracted with pre-normalization to the sample protein content (500 μg/mL), while serum samples were extracted without pre-normalization to the sample protein content. Global metabolomics profiling was performed on a Thermo Q-Exactive Oribtrap mass spectrometer with Dionex UHPLC and autosampler. All samples were analyzed in positive and negative heated electrospray ionization with a mass resolution of 35,000 at $m/z$ 200 as separate injections. Separation was achieved on an ACE 18-pfp 100 × 2.1 mm, 2 μm column with mobile phase A as 0.1% formic acid in water and mobile phase B as acetonitrile. This is a polar embedded stationary phase that provides comprehensive coverage, but does have some limitation is the coverage of very polar species. The flow rate was 350 μL/min with a column temperature of 25 °C. 4 μL was injected for negative ions and 2 μL for positive ions. MZmine (freeware) was used to identify features, deisotope, align features and perform gap filling to fill in any features that may have been missed in the first alignment algorithm. All adducts and complexes were identified and removed from the data set. A total of 1035 and 888 compounds were detected in the negative and positive phase respectively in skeletal muscle, 2686 and 2636 compounds were detected in the negative and positive phase respectively in serum samples and 764 and 696 compounds were detected in the negative and positive phase, respectively, in tumor samples. Unknown features were excluded from subsequent analyses. Statistical analyses were conducted using MetaboAnalyst online software (version 5.0, https://www.metaboanalyst.ca/home.xhtml). All data were normalized (normalization by the sum) and scaled (Pareto scaling). T-tests were performed for each experimental group against its appropriate Sham group. Metabolite abundance was deemed significantly altered if it presented an absolute fold-change greater than 1.2 and a $p < 0.05$.

**Statistics and reproducibility**. Data normality was tested with Shapiro-Wilk test and parametric or non-parametric tests were used. Separate t-tests were conducted to examine the effect of KPC tumor burden on body and tissue masses. T-tests were also performed to evaluate the effect of tumor burden on twitch and tetanic force in the WT mice, while one-way ANOVAs were used to examine this effect in the MuRF1[-/-] mice. One-way ANOVAs were also used to compare tumor mass across genotype as well as body mass and tissue mass throughout the progression of cancer in WT mice. Two-way ANOVAs were performed to examine the effect of tumor burden on fiber type distribution and force-frequency relationship. When statistically significant differences were detected by ANOVAs, Tukey (for one-way ANOVA), Dunn's (for Kruskal–Wallis one-way ANOVA), and Sidak (two-way ANOVA) post hoc analyses were used to test for differences among pairs of means. For, skeletal muscle fiber cross-sectional area, data were binned, fit with a Gaussian least squares regression and significance was determined by calculating the extra sum-of-squares $F$-test. An exponential one-phase decay model was used to compare fatigue resistance. Survival data were analyzed with the Kaplan–Meier method using a Mantel-Cox test. The alpha level for significance was set to 0.05 and data are expressed as mean ± standard error. All effect size values (i.e., $\eta^2$, $\omega^2$, and $R^2$) reported were obtained with Prism (version 9.3.1, GraphPad Software, La Jolla, CA, USA), with the exception of effect sizes associated with Kruskal–Wallis analyses. For these analyses, the following formula was used: $\eta^2 = (H – k + 1)/(n – k)$ where $H$ is the Kruskal–Wallis statistic, $k$ is the number of groups and $n$ is the total number of observations. Except for the metabolomics data, all statistical analyses were performed with Prism (version 9.3.1, GraphPad Software, La Jolla, CA, USA). All sample sizes are provided in the figures and figure legends.

**Reporting summary**. Further information on research design is available in the Nature Portfolio Reporting Summary linked to this article.

# Data availability

The authors declare that all data generated or analyzed in this study are either (i) deposited to the the PRIDE repository or (ii) included in this published article and its supplementary information files. The global proteome and ubiquitinome respective datasets are accessible with the identifiers PXD039158 and PXD039151. The data used to generate figures are provided in Supplementary Data 11.

# Code availability

No custom computer code or algorithm was used for this manuscript.

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

## Acknowledgements

The authors would like to thank the Physiological Assessment Core directed by Dr. Elisabeth Barton at the University of Florida, which was established under the Senator Paul D. Wellstone Muscular Dystrophy Cooperative Research Center awarded to H. Lee Sweeney (overall PI; U54AR052646), for performing the muscle contractility experiments; the University of Florida Southeast Center for Integrated Metabolomics, for performing all metabolomics experiments and assisting with data analyses; Tongjun Gu and the Bioinformatics Core from the Interdisciplinary Center for Biotechnology Research of the University of Florida for their insightful discussion of the proteomics bioinformatic analyses; and Dr. Matthew P. Stokes and Cell Signaling Technology for performing all proteomics experiments and assisting with analyses and interpretation of the results. This work was supported by the National Institute of Arthritis, Musculoskeletal and Skin Diseases (R01AR060209-08S2 to A.R.J.), the University of Florida Office of Research Seed Fund (to A.R.J.), the Clinical and Translational Science Institute of the University of Florida (to D.N.). D.N. was supported by the Swiss National Science Foundation (P4P4PM_191137).

## Author contributions

D.N., J.D., S.M.J., and A.R.J. conceived the study; D.N., O.L., A.D., R.E.S., J.Z.S., G.A.W., J.D., S.M.J., and A.R.J. participated in data acquisition, analysis and interpretation; C.L., D.W.H., and H.L.S. contributed to data analysis and interpretation; D.N., S.M.J., and A.R.J. wrote the manuscript. All authors read and approved the submitted version and have agreed to be personally accountable for the author's own contributions and to ensure that questions related to the accuracy or integrity of any part of the work are appropriately investigated and resolved.

## Competing interests

The authors declare no competing interests.
