## [Peer Review File · Communications Biology]

Reviewers' comments:

Reviewer #1 (Remarks to the Author):

Cachexia is a debilitating disease that results in decreased quality of life in PDAC patients and in many cases is the cause of mortality more than the cancer itself. Research in this field has not led to any definite markers of cachexia, however the increased muscle loss has been attributed to increased expression of muscle-specific E3 ligases MURF1 and Atrogin1. In this study "Blocking muscle wasting via deletion of the muscle-specific E3 ubiquitin ligase MuRF1 impedes pancreatic tumor growth" authors have solidified the role of MURF1 in tumor-induced cachexia and have shown how murf1 knockout mice could circumvent muscle and fat loss otherwise induced by tumor burden. Slow tumor growth and increased overall survival in knockout mice and the proteomic and metabolomic studies confirm the role of MURF1 in cancer-induced cachexia. Proper controls have been used and different pathways studied to complement results from each pathway. This study will add to the field of cachexia research and pave the way to considering MURF1 as a drug target for treatment of cachexia.

Reviewer #2 (Remarks to the Author):

The manuscript by Neyroud et al investigated the requirement of the muscle-specific E3 ubiquitin ligase, MuRF1, for muscle wasting induced by pancreatic cancer. They found that KPC tumors induced progressive wasting of skeletal muscle and systemic metabolic reprogramming in WT mice, but not MuRF1^{-/-} mice. KPC tumors from MuRF1^{-/-} mice also grew slower, and showed an accumulation of metabolites normally depleted by rapidly growing tumors. They validated in female mice that MuRF1 deletion also protects against KPC T42D-induced wasting, slows tumor growth and extends survival. Overall, this study is interesting and novel. It provides comprehensive analysis of the metabolome and ubiquitination-proteasome alteration in WT and MuRF1^{-/-} mice bearing tumors. I have the following comments.

1. Besides MuRF1, several other proteins are also critical for muscle wasting, such as Atrogin-1 and UBR2. Why would they focus on MuRF1? Please clarify.
2. For the results of metabolome analysis, as shown in Table 1-2, they compared WT and MuRF1^{-/-} mice bearing tumors. Did they have any chance to examine the metabolites of normal pancreas, muscle and serum in Sham mice?
3. They validated their findings in female mice in Fig. 7. Are female mice showing similar patterns of metabolome and ubiquitination-proteasome alteration?
4. In the figure legend of Fig. 1, the authors stated that "MuRF1 (Trim63) is increased in skeletal muscle of tumor-bearing hosts.". Although previous studies have demonstrated it, in Fig. 1, they did not examine MuRF1 expression in skeletal muscle of tumor-bearing hosts. Please rephrase it.
5. Please add the statistics analysis of Fig. 7A-7B.
6. Line 160-168, font types are inconsistent.

Reviewer #3 (Remarks to the Author):

In this manuscript, Neyroud et al investigated the role of MuRF1, a muscle-specific E3 ubiquitin ligase, impact of muscle wasting induced by pancreatic cancer. Using an orthotopic model of pancreatic cancer (either WT or KO MuRF1 transgenic mouse models), the authors demonstrated that MuRF1 is necessary for the increases in cytoskeletal and muscle contractile protein ubiquitination as well as for the inhibition of pathways that support protein synthesis. In addition, deletion of Murf1 can reprogram

the systemic and tumor metabolism, delaying tumor growth.

This is an interesting and well-organized study that unveils the function of MuRF1 in controlling skeletal muscle wasting in pancreatic cancer, establishing that directly interfering with muscle wasting can impact the tumor metabolism. The only major weakness stands in the fact that having in hands such a remarkable and robust effect on systemic and tumor metabolism, one would have expected to see a further validation of the effects induced by the metabolites that showed altered abundance across tissues, as reported in Figure 6. The authors may include additional experiments validating how MuRF1 loss deprives tumor cells from key energy metabolites and reduces tumor growth and expanding the discussion session on this topic, in order to implement the potential clinical impact of their findings

Responses to Reviewers' comments

Reviewer #1

Comment: "Cachexia is a debilitating disease that results in decreased quality of life in PDAC patients and in many case is the cause of mortality than the cancer itself. research in this field has not lead to any definite markers of cachexia, however the increased muscle loss has been attributed to increased expression of muscle specific E3 ligases MURF1 and Atrogin1. in this study "Blocking muscle wasting via deletion of the muscle specific E3 ubiquitin ligase MuRF1 impedes pancreatic tumor growth" authors have solidified the role of MURF 1 in tumor induced cachexia and have showed how murf1 knockout mice could circumvent muscle and fat loss otherwise induced by tumor burden. slow tumor growth and increased overall survival in knockout mice and the proteomic and metabolomic studies confirm the role of MURF 1 in cancer induced cachexia. proper controls have been used and different pathways studied to compliment results from each pathway. this study will add to the field of cachexia research and pave way to considering MURF1 as a drug target for treatment or cachexia."

Response: We would like to thank this reviewer for her/his review of our manuscript.

Reviewer #2

Comment 1: "The manuscript by Neyroud et al investigated the requirement of the muscle-specific E3 ubiquitin ligase, MuRF1, for muscle wasting induced by pancreatic cancer. They found that KPC tumors induced progressive wasting of skeletal muscle and systemic metabolic reprogramming in WT mice, but not MuRF1^{-/-} mice. KPC tumors from MuRF1^{-/-} mice also grew slower, and showed an accumulation of metabolites normally depleted by rapidly growing tumors. They validated in female mice that MuRF1 deletion also protects against KPC T42D-induced wasting, slows tumor growth and extends survival. Overall, this study is interesting and novel. It provides comprehensive analysis of the metabolome and ubiquitination-proteasome alteration in WT and MuRF1^{-/-} mice bearing tumors. I have the following comments."

Response 1: We would like to thank this reviewer for her/his review of our manuscript.

Comment 2: "Besides MuRF1, several other proteins are also critical for muscle wasting, such as Atrogin-1 and UBR2. Why would they focus on MuRF1? Please clarify."

Response 2: We fully acknowledge that there are additional ubiquitin ligases in skeletal muscle that play important roles in muscle wasting, including Ubr2, which has also been identified as a key player in cancer-induced muscle loss (Gao et al., PNAS 2022). Nonetheless, MuRF1 is consistently elevated in skeletal muscles of mice and people with cancer who exhibit cachexia (ref), providing strong rationale to investigate the role of MuRF1 in cancer-induced muscle loss. Conducting cancer cachexia studies utilizing mice lacking other ubiquitin ligases in skeletal muscle, such as atrogin-1 and Ubr2, and conducting similar omics analyses (proteomics, ubiquitinomics, metabolomics) as performed in the current study, was simply beyond the scope of the current study.

To acknowledge the important role of the E3 ligase, Ubr2, in muscle wasting induced by cancer, we have now added the following sentences within the introduction (l. 63-67), and

within the results/discussion (l. 225-227) to acknowledge the important findings from this study which revealed Ubr2 as a key E3 ligase that ubiquitylates MYH4 and MYH1 proteins in the context of cancer, leading to loss of muscle mass and force production.

- L. 63-67: "In the context of muscle wasting, various E3 ligases have been shown to be involved in contractile protein degradation (see ¹⁷⁻¹⁹ for review), including the muscle-specific E3 ligases – F-Box Protein 32 (*Fbxo32/atrogin-1*) ^{17,20-22} and muscle RING finger protein 1 (*MuRF1/Trim63*) ^{17,20,23-28} – as well as the more ubiquitously expressed E3 ligase, ubiquitin protein ligase E3 component N-Recognin 2 (*Ubr2*) ^{21,29}."
- L. 225-227: "Notably, a recent study demonstrated that MYH1 and MYH4 proteins are also ubiquitinated by another E3 ligase, UBR2, whose muscle-specific deletion prevented fast-twitch muscle wasting in response to tumor burden ²⁹."

Comment 3: "For the results of metabolome analysis, as shown in Table 1-2, they compared WT and MuRF1-/- mice bearing tumors. Did they have any chance to examine the metabolites of normal pancreas, muscle and serum in Sham mice?"

Response 3: We have not examined normal pancreas metabolome in this study as our main goal was to determine if the slowed tumor growth observed in the absence of MuRF1 (i.e. a skeletal muscle E3-ubiquitin ligase) was mediated by an alteration in tumor metabolome. With respect to muscle and serum, we examined their metabolome in Sham mice. The data presented in Table 2 and 3 are indeed fold changes vs. Sham. The absolute data for the Sham mice (as well as tumor-bearing mice) can be found in Supplementary file 5, 7 and 8).

Comment 4: "They validated their findings in female mice in Fig. 7. Are female mice showing similar patterns of metabolome and ubiquitination-proteasome alteration?"

Response 4: While performing metabolomic or ubiquitinome-proteomic analyses in tissues from both male and female mice could be insightful, both sexes showed comparable effects of MuRF1 deletion on the major outcomes, including protection against tumor-induced muscle loss and slowed tumor growth. and. Because sex-differences were not apparent in the major cancer and cachexia outcomes, there was not strong justification for duplicating the omics studies in both sexes, which would double the cost. We therefore limited the omics studies to a single sex.

"Although *omics* analyses were not conducted on tissues from these mice, these findings in female mice, using a different KPC cell line and conducted in a different lab, lend strong support for the role of MuRF1 in mediating cancer-associated muscle wasting and tumor growth." (l. 348-350).

Comment 5: "In the figure legend of Fig. 1, the authors stated that "MuRF1 (Trim63) is increased in skeletal muscle of tumor-bearing hosts.". Although previous studies have demonstrated it, in Fig. 1, they did not examine MuRF1 expression in skeletal muscle of tumor-bearing hosts. Please rephrase it."

Response 5: We thank the reviewer for catching this oversight. We have modified Fig. 1 title as follow: "MuRF1 (Trim63) deletion protects against muscle wasting, skeletal muscle fiber atrophy and slows tumor growth."

Comment 6: “Please add the statistics analysis of Fig. 7A-7B.”

Response 6: Statistics have been added to panel A. For panel B, no statistics were conducted due to the different number of mice at different time points.

Comment 7: “Line 160-168, font types are inconsistent.”

Response 7: We thank the reviewer for his/her thorough review of the manuscript and catching this inconsistency. This font inconsistency has been resolved.

Reviewer #3

Comment: “In this manuscript, Neyroud et al investigated the role of MuRF1, a muscle-specific E3 ubiquitin ligase, impact of muscle wasting induce by pancreatic cancer. Using an orthotopic model of pancreatic cancer (either WT or KO MuRF1 transgenic mouse models), the authors demonstrated that MuRF1 is necessary for the increases in cytoskeletal and muscle contractile protein ubiquitination as well as for the inhibition of pathways that support protein synthesis. In addition, deletion of Murf1 can reprogram the systemic and tumor metabolism, delaying tumor growth.

This is an interesting and well-organized study that unveils the function of MuRF1 in controlling skeletal muscle wasting in pancreatic cancer, establishing that directly interfering with muscle wasting can impact the tumor metabolism. The only major weakness stands in the fact that having in hands such a remarkable and robust effect on systemic and tumor metabolism, one would have expected to see a further validation of the effects induced by the metabolites that showed altered abundance across tissues, as reported in Figure 6. The authors may include additional experiments validating how MuRF1 loss deprives tumor cells from key energy metabolites and reduces tumor growth and expanding the discussion session on this topic, in order to implement the potential clinical impact of their findings.”

Response: We would like to thank this reviewer for her/his review of our manuscript. We agree with the reviewer’s assessment that there is a remarkable and robust impact of MuRF1 KO on systemic and tumor metabolism. However, studies investigating the direct effects of specific metabolites regulated by MuRF1 on *in vivo* tumor metabolism, tumor growth and cachexia are not trivial, and would require extensive studies beyond the scope of the current study.

However, we have now incorporated a sentence to acknowledge that additional validation studies are needed to investigate the key metabolites regulated by MuRF1 on *in vivo* metabolism, tumor growth and cachexia: “These results pave the way for future validation studies investigating the direct effects of specific MuRF1-regulated metabolites on *in vivo* tumor growth.” (l. 349-351).

REVIEWERS' COMMENTS:

Reviewer #2 (Remarks to the Author):

The authors have adequately addressed all my concerns, no more questions.

Reviewer #3 (Remarks to the Author):

The authors addressed my comments.